# TgAP2IX-5 is a key transcriptional regulator of the asexual cell cycle division in *Toxoplasma gondii*

Asma S. Khelifa[1], Cecilia Guillen Sanchez [1], Kevin M. Lesage[1], Ludovic Huot[1], Thomas Mouveaux[1], Pierre Pericard[2], Nicolas Barois[1], Helene Touzet[2,3], Guillemette Marot[2,4], Emmanuel Roger[1] & Mathieu Gissot [1]✉

Apicomplexan parasites have evolved efficient and distinctive strategies for intracellular replication where the timing of emergence of the daughter cells (budding) is a decisive element. However, the molecular mechanisms that provide the proper timing of parasite budding remain unknown. Using *Toxoplasma gondii* as a model Apicomplexan, we identified a master regulator that controls the timing of the budding process. We show that an ApiAP2 transcription factor, TgAP2IX-5, controls cell cycle events downstream of centrosome duplication. TgAP2IX-5 binds to the promoter of hundreds of genes and controls the activation of the budding-specific cell cycle expression program. TgAP2IX-5 regulates the expression of specific transcription factors that are necessary for the completion of the budding cycle. Moreover, TgAP2IX-5 acts as a limiting factor that ensures that asexual proliferation continues by promoting the inhibition of the differentiation pathway. Therefore, TgAP2IX-5 is a master regulator that controls both cell cycle and developmental pathways.

[1] Univ. Lille, CNRS, Inserm, CHU Lille, Institut Pasteur de Lille, U1019-UMR 9017-CIIL-Center for Infection and Immunity of Lille, F-59000 Lille, France. [2] Univ. Lille, CNRS, Inserm, CHU Lille, Institut Pasteur de Lille, US 41-UMS 2014-PLBS, bilille, F-59000 Lille, France. [3] Univ. Lille, CNRS, Centrale Lille, UMR 9189-CRIStAL-Centre de Recherche en Informatique Signal et Automatique de Lille, F-59000 Lille, France. [4] Univ. Lille, Inria, CHU Lille, ULR 2694-METRICS: Evaluation des technologies de santé et des pratiques médicales, F-59000 Lille, France. ✉email: mathieu.gissot@pasteur-lille.fr

Apicomplexa is a phylum consisting of unicellular, obligate, intracellular protozoan parasites, which includes various human pathogen species, such as *Plasmodium* spp. (causative agent of malaria), *Toxoplasma* (cause of toxoplasmosis), and *Cryptosporidium* spp. (cause of cryptosporidiosis).

Apicomplexan parasites trigger disease associated with an uncontrollable expansion of parasite biomass resulting in inflammation and host-cell destruction[1]. Although, apicomplexan parasites present a sexual cycle within the definitive host, the pathogenesis of these parasites results from the ongoing asexual replication cycles within the host's cells. All Apicomplexa possess complex life cycles consisting of parasite propagation controlled by the tight regulation of the cell cycle and result in the formation of new daughter cells containing one nucleus and a complete set of organelles[2]. Apicomplexa have evolved the ability to independently divide their nucleus (nuclear cycle) and produce the parasite body through a process that is termed budding (budding cycle). This allows flexibility in order to produce a suitable number of offspring in a single cell cycle while allowing the parasite to cope with the different host-cell environments[3]. Apicomplexa have evolved efficient and distinctive strategies for intracellular replication[1]. How the division pattern is chosen to ensure parasite expansion during host-cell infection remains unanswered. However, it has been clear that the mode of division is dependent on the timing and coordination of the nuclear and budding cycles[4,5].

*Plasmodium falciparum* and *Toxoplasma gondii* represent two modern branch points of the Apicomplexa phylum, a divergence that occurred several hundred million years ago[6]. This divergence has led to changes in the usage of different cell division patterns. For instance, in its intermediate host, *P. falciparum* replicates through schizogony, a division pattern, which produces a multinucleated intermediate (schizont) where the daughter parasites bud from the periphery at the closure of the division process to produce infective merozoites. By contrast, in its intermediate hosts, *T. gondii* undergoes endodyogeny, a division pattern that consists of the formation of two daughter parasites within the mother cell. In this case, the formation of new daughter cells occurs within the cytoplasm rather than from the periphery[1] and is coordinated to arise simultaneously with nuclear division. In addition to endodyogeny, which is the simplest form of internal budding, *T. gondii* asexually divides within its definitive host through a division scheme that closely resembles schizogony and is known as endopolygeny. It consists of the production of multiple nuclei and a final step of daughter cell formation where parasites perform internal budding in the cytoplasm, unlike schizogony[7]. Furthermore, *Babesia spp.* divide by a binary form of schizogony, where nuclear and budding cycle are linked, with the difference that they bud from the periphery rather than internally[8]. These distinct replication patterns (endodyogeny, schizogony, and endopolygeny) rely on the coordination of the timing of the budding and nuclear cycles. Apicomplexa, such as *T. gondii*, which exhibit endodyogeny and endopolygeny, are able to employ several division patterns depending on the cellular microenvironment and the developmental stage of the parasite's lifecycle. Such a flexibility suggests overlapping mechanisms in control of the cell cycle independent of the division pattern used.

Apicomplexa show a peculiar cell cycle with a closed mitosis that is divided into three phases (G1, S, and M) while the G2 phase is apparently absent[9]. Centrosomes play a central role in controlling the cell cycle in Apicomplexa. Division and maturation of the centrosome controls the progression and the coordination of the nuclear and budding cycle[5,9–12]. Evidence suggests that soon after centrosome division, centrosome maturation through kinases is key to the activation of daughter cell formation[10,11].

During the Apicomplexa cell cycle, division, and segregation of organelles is a highly ordinated process in order to ensure that each daughter parasite acquires the proper complement of organelles[13]. This highly controlled process implies a tight regulation of the gene expression during the cell cycle. This was illustrated by the transcriptomic analysis of synchronous cell cycle populations indicating that *T. gondii* may have adapted a "just in time" mode of expression whereby transcripts and proteins are produced right when their function is needed[14]. This suggests that transcriptional regulation of gene expression may exert a centralized control on both centrosome activation and the expression of proteins needed for daughter cell formation.

However, little is known on the potential regulators that may control this transcriptional switch. Apicomplexan genomes encode a family of putative transcription factors (TFs) that are characterized by the possession of one or more DNA binding AP2 domains[15]. ApiAP2 transcription factors were shown to control expression profiles during *T. gondii* differentiation[16–18]. ApiAP2s were also shown to cooperatively control the expression of virulence factors[19,20] in *T. gondii* indicating that this family of transcription factors may control cell-cycle-dependent expression profiles as also suggested for *P. berghei*[21]. Moreover, *T. gondii* ApiAP2s were shown to bind to promoters and regulate cell-cycle-specific activity, indicating that they play an active role in regulating cell-cycle-specific expression patterns in this parasite[19,20]. In this study, we functionally characterize a cell-cycle-dependent ApiAP2 transcription factor, TgAP2IX-5. We show that it controls cell cycle events downstream of centrosome duplication including organelle division and segregation. TgAP2IX-5 binds to the promoter of hundreds of genes and controls the activation of the S/M-specific cell cycle expression program. Reversible destabilization of the TgAP2IX-5 protein showed that its expression is sufficient for activation of the budding cycle in multinucleated parasites. Therefore, controlled TgAP2IX-5 expression allows for the switching from endodyogeny to endopolygeny division patterns. TgAP2IX-5 acts as a master regulator controlling the budding cycle and therefore, asexual cell cycle division patterns in this parasite.

## Results

**TgAP2IX-5 is a cell-cycle-regulated transcription factor essential for parasite growth and proliferation.** To find specific regulators of the cell-cycle-dependent expression program, we identified potential transcriptional regulators whose expression was cell cycle regulated[14]. Among these regulators, we focused on the ApiAP2 family of transcription factors and discovered that the *tgap2ix-5* transcript was dynamically expressed with a peak of expression within the S phase (toxodb.org). TgAP2IX-5 AP2 domain is conserved among apicomlexan parasites (eupathdb.org). To confirm that the TgAP2IX-5 protein was also cell cycle regulated, we produced a *T. gondii* strain that had an epitope-tagged version of this gene at its endogenous locus. Immunofluorescence assays using cell cycle markers demonstrated that TgAP2IX-5 is a nuclear protein mostly expressed during the early S phase (Fig. 1a). TgAP2IX-5 expression was detected at the G1/S transition and early S phase when centrosomes are divided but remained in close proximity of each other (as identified by TgCentrin1; a marker of the outer core of the centrosome, Fig. 1a upper panel, Supplementary Fig. 1a). TgAP2IX-5 was detected before centromere division (as identified by TgChromo1; a marker of pericentromeric chromatin[22], Fig. 1a middle panel, Supplementary Fig. 1b–c) but not after (late S phase), indicating that TgAP2IX-5 was only present during the G1/S transition and early S phase. TgAP2IX-5 was no longer detected when parasites enter the S/M phase (as indicated by an early budding marker

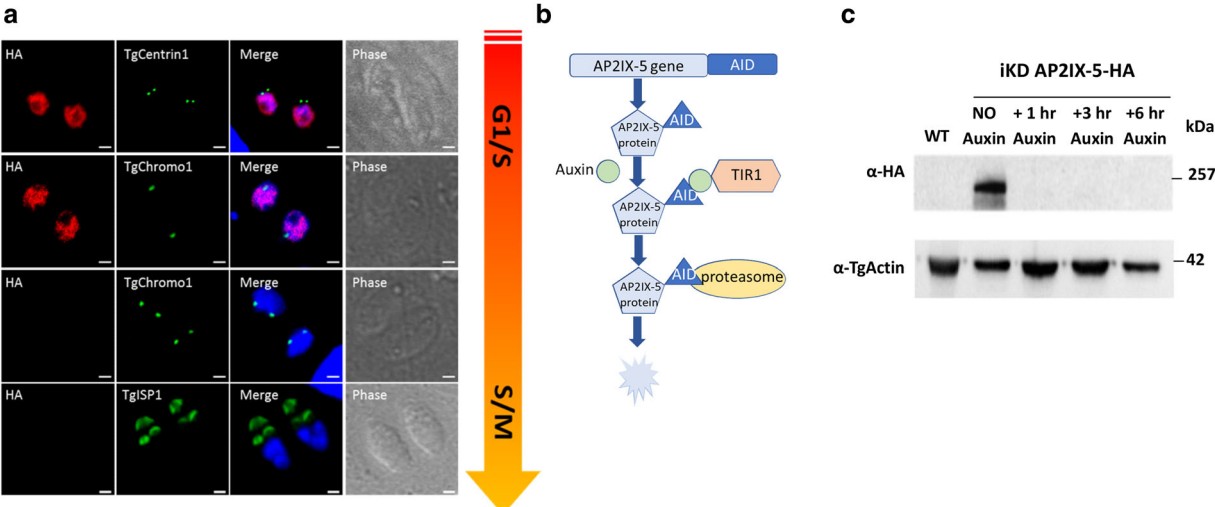

**Fig. 1 TgAP2IX-5 is a cell-cycle-regulated protein. a** Confocal imaging demonstrating the expression of the TgAP2IX-5 protein using anti-HA antibody during the tachyzoite cell cycle. Anti-TgCentrin1, anti-TgChromo1, and anti-TgISP1 were used as cell cycle markers. The schematic cell cycle phase is indicated on the right side of the figure and the scale bar (1 μm) is indicated on the lower right side of each confocal image. **b** Schematic representation of the AID system used to degrade the TgAP2IX-5 protein. This system consists of introducing a recognition sequence into the gene of interest, which expresses a protein fused to the recognition sequence and in the presence of Auxin, will be recognized by the TIR1 protein and degraded by the proteasome. This system was used to produce an inducible knockdown parasite in which the expression of TgAP2IX-5 can be controlled. **c** Western blot of total protein extract from the parental and iKD TgAP2IX-5 strains treated with Auxin for different durations of time validating the AID system. Western blots were probed with anti-HA to detect the presence of TgAP2IX-5 protein (upper panel), anti-TgActin was used as a control for normalization (lower panel).

TgISP1, (Fig. 1a, lower panel)). Therefore, TgAP2IX-5 protein expression is tightly controlled during the cell cycle.

To identify the biological role of TgAP2IX-5, an inducible knockdown (iKD) mutant of TgAP2IX-5 was generated using the AID system, which allowed for the conditional depletion of the TgAP2IX-5 protein (Fig. 1b). For this purpose, we produced a strain that presented an AID-HA insert at the 3' end of the *TgAP2IX-5* gene using a CrispR/Cas9 strategy (Supplementary Fig. 2a). The correct locus of integration of this insert was confirmed by PCR (Supplementary Fig. 2b). The AID system allows for the conditional depletion of the target protein after addition of auxin (indoleacetic acid) (Fig. 1b). A single detectable band at the expected protein size (251 kDa) was present in absence of auxin in the iKD TgAP2IX-5 strain as verified by Western Blot (Fig. 1c). Rapid depletion of the TgAP2IX-5 protein (as early as 1 h) was obtained in iKD TgAP2IX-5 mutant strain grown in media containing auxin (Fig. 1c).

**iKD TgAP2IX-5 mutant displays a defect in daughter parasite formation.** To phenotypically characterize the iKD TgAP2IX-5 mutant, a standard growth assay was carried out. The growth capacity of the iKD TgAP2IX-5 mutant in the presence of auxin was tested and we observed that the iKD TgAP2IX-5 mutant growth ability was drastically decreased in the presence of auxin (Fig. 2a). In fact, growth was completely abrogated in the iKD TgAP2IX-5 mutant in presence of auxin with a calculated mean of 1.22 parasites per vacuole while the parental strain (in presence and absence of auxin) and the iKD TgAP2IX-5 mutant in absence of auxin grew at a mean of 2.5 parasites per vacuole. This demonstrated a complete blockage of the proliferation capacity of the parasite in the absence of TgAP2IX-5 (Fig. 2a). A plaque assay, that measured the ability of the parasite to grow and invade over a period of 7 days confirmed the phenotype with the absence of lysis plaques in the iKD TgAP2IX-5 strain treated with auxin

contrary to the control used (parental strain in presence of auxin) where lysis plaques were observed (Supplementary Fig. 3a). These results suggest that TgAP2IX-5 expression is essential for the parasite's growth and proliferation.

To better assess the growth phenotype, we examined the parasite by IFA using a nucleus marker (TgENO2) and an inner membrane complex (IMC) marker (TgIMC1), a network of flattened vesicles lying beneath the plasma membrane (Fig. 2b). It was apparent that the parasite accumulated nuclei and did not form daughter cells in absence of TgAP2IX-5 while exhibiting normal daughter cell formation in presence of the protein. To confirm the inability of the parasite to form daughter cells in presence of auxin, we performed transmission electron microscopy and observed that in absence of auxin the iKD TgAP2IX-5 mutant was able to produce daughter cells, while in presence of auxin, the parasite accumulated nuclei associated with lack of apparent daughter cell formation (Fig. 2c). To confirm these observations, we measured the number of nuclei per parasite (Fig. 2d and Supplementary Fig. 3b–c). Depending on the timing during the cell cycle, the parasites exhibit either one nucleus (80% of the total parasite population) or two nuclei (20% of the total parasite population) (before or after cytokinesis) for the parental strain (in absence or in presence of auxin) and for the iKD TgAP2IX-5 strain in absence of auxin (Fig. 2d and Supplementary Fig. 3b–c). By contrast, the percentage of parasite exhibiting more than 2 nuclei increased overtime in the iKD TgAP2IX-5 strain in presence of auxin (Fig. 2d and Supplementary Fig. 3b–c). The observation of multinucleated parasites in presence of auxin suggests that the iKD TgAP2IX-5 mutant displays a defect in daughter parasite formation but not in the ability of the parasite to multiply and segregate its nuclear material. To assess the capacity of the iKD TgAP2IX-5 mutant to produce daughter cells, we recorded the number of parasite cells that were in the process of producing daughter cells through internal budding (Fig. 2b, e and Supplementary Fig. 4a–b), therefore measuring the parasite's

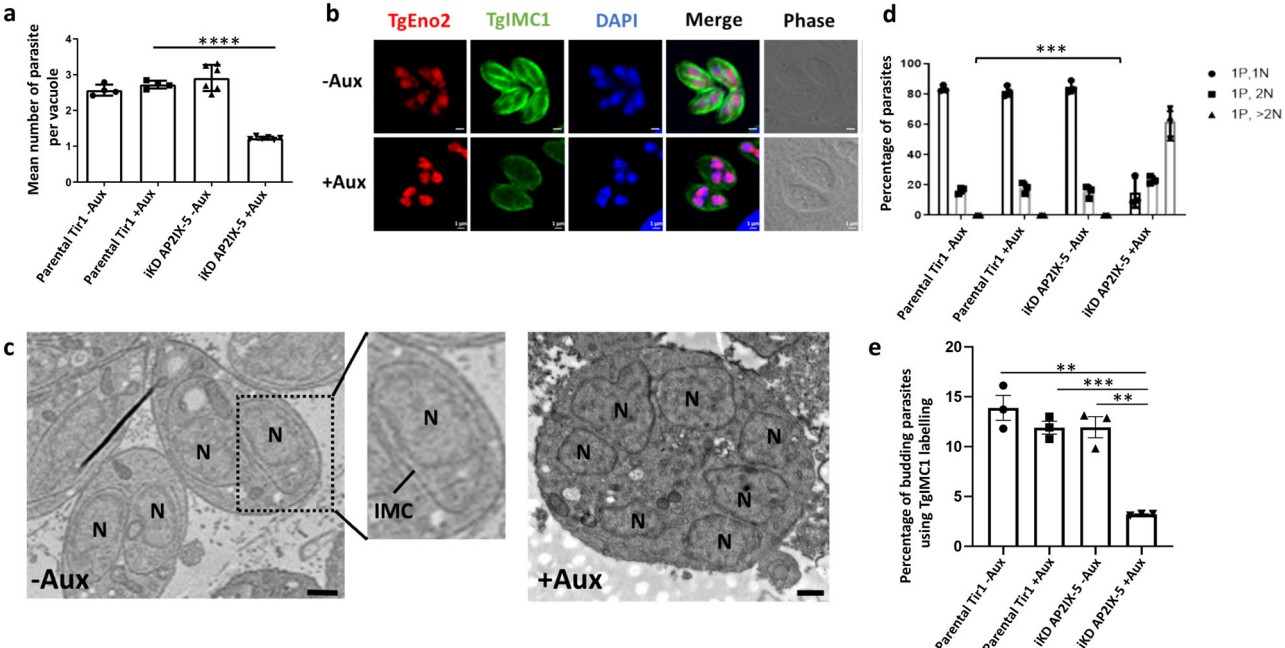

**Fig. 2 Characterization of TgAP2IX-5 iKD parasite. a** Growth assay for parental and iKD TgAP2IX-5 strains in the absence and presence of auxin treatment for 24 h. The number of parasites per vacuole was measured and the average number of parasites is represented within the graph. A total of 100 vacuoles were counted for each replicate. A Student's *t*-test was performed; two-tailed *p*-value: ****$p < 0.0001$; mean ± s.d. ($n = 4$ independent experiments). **b** Confocal imaging of iKD TgAP2IX-5 labelled with TgEno2 (red) and TgIMC1(green) in the presence and absence of auxin treatment. Auxin was added 24 h post-infection. DAPI was used to stain the nucleus. Scale bar is indicated at the lower right side of each image. **c** Electron microscopy scans demonstrating the structural morphology of the nucleus in iKD TgAP2IX-5 parasite in the absence and presence of auxin. (N) represents the nucleus. Two daughter parasites are formed within each mother parasite in absence of auxin. Multinucleated parasites are visible in presence of auxin. Scale bar (500 nm) is indicated at the lower right side in the TEM images. **d** Bar graphs representing nucleus per parasite counts for parental and iKD TgAP2IX-5 strains in the absence and presence of auxin (12 h treatment). P stands for parasites and N stands for nuclei. A Student's *t*-test was performed comparing mean percentage of multinucleated parasite between the control (Parental in absence of auxin) and iKD TgAP2IX-5 in the presence of auxin, two-tailed *p*-values: ***$p = 0,0004$; mean ± s.d. ($n = 3$ independent experiments). **e** Bar graph representing the percentage of daughter parasite formation in the absence and presence of 6 h of auxin treatment using TgIMC1 labelling, A Student's *t*-test was performed, two-tailed *p*-values: ***$p = 0.0002$, **$p = 0.0012$; mean ± s.d. ($n = 3$ independent experiments).

ability to produce daughter cells at an early stage of internal budding. Using two different markers, we were able to show a drastic decrease of the number of parasites undergoing budding in the iKD TgAP2IX-5 strain after 6 h of auxin treatment (Fig. 2e and Supplementary Fig. 4a–b). These results indicate that TgAP2IX-5 plays an important role during the early stages of internal budding and is necessary for daughter cell formation.

**TgAP2IX-5 is required at a precise timepoint of the cell cycle**. The *T. gondii* cell cycle has been described precisely and presents a well-organized timeline for the division of the subcellular structures of the parasite[13] establishing the following sequence of organelle duplication and segregation: first the centrosome is duplicated and divided and then the Golgi complex, the apicoplast, the nucleus, the cytoskeleton (e.g., the IMC), the endoplasmic reticulum and eventually the mitochondrion. To identify the exact timepoint at which TgAP2IX-5 affects daughter parasite formation, a study of the effect of TgAP2IX-5 on organelle duplication and segregation was carried out.

During division, the centrosome (outer core) divides first and then the centromere (as well as the inner core centrosome) follows. We measured the ability of the centrosome and centromere to divide within the iKD TgAP2IX-5 strain using TgCentrin1 and TgChromo1 as a centrosome and centromere marker, respectively. For the parental and the iKD TgAP2IX-5 strains, the centrosome to nucleus and centromere to nucleus

ratio were recorded after 6 hours of auxin treatment (Fig. 3a–c). The recorded ratios of centrosome to nucleus and centromere to nucleus are close to 1 in both the presence and absence of auxin (Fig. 3b–c) despite that the Student's *t*-test carried out for statistical analysis shows a significantly lower number for both centrosome to nucleus and centromere to nucleus ratios. These results suggest that TgAP2IX-5 does not have a drastic effect on the replication of the centrosome. Similarly, the centromere division is minimally affected, a result that is in line with the multiplication of nuclei observed in the iKD TgAP2IX-5 strain in presence of auxin, as confirmed by the labelling of TgSFA2, another centrosome marker (Supplementary Fig. 5a–b).

Since centrosome division remained minimally affected by TgAP2IX-5, a study of the proceeding organelles to divide in the *T. gondii* organellar cell cycle division timeline was carried out (Fig. 4a). The Golgi complex was labelled in parasites and the ratio of Golgi to nucleus was calculated (Fig. 4b). We observed no significant difference between the ratios of Golgi to nucleus in the presence or absence of auxin, therefore suggesting that TgAP2IX-5 depletion does not impact Golgi division and segregation (Fig. 4c). We then observed plastid division and segregation (Fig. 4b) and measured the plastid to nucleus ratio. This ratio is significantly lower in the iKD TgAP2IX-5 strain in presence of auxin (Fig. 4d), suggesting that plastid division is blocked in the mutant. The plastid division is a multistep process where it first elongates before the completion of scission and segregation[23]. To determine the exact timepoint at which plastid division is

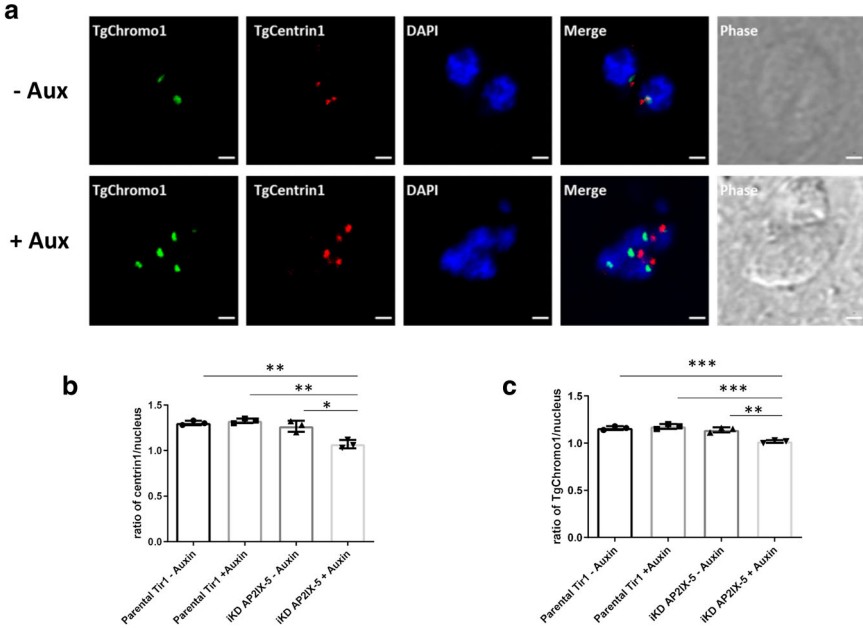

**Fig. 3 Centrosome division is mostly unaffected in the absence of TgAP2IX-5. a** Confocal imaging of iKD TgAP2IX-5 labelled with TgChromo1 and TgCentrin1. TgChromo1 is indicated in green. TgCentrin1 is indicated in red. DAPI was used to stain the nucleus. Scale bar is indicated at the lower right side of each image. Occasional disconnection between centromere and outer-centrosome was observed but does not represent the majority of cases. **b** Bar graph representing TgCentrin1: nucleus ratio using the parental and iKD TgAP2IX-5 strains in the absence and presence of auxin treatment for 6 h. A Student's t-test was performed, two-tailed p-values: **$p = 0.0015$ (iKD TgAP2IX-5 +auxin compared to Parental Tir1 -auxin), **$p = 0.001$ (iKD TgAP2IX-5 +auxin compared to Parental Tir1 +auxin), **$p = 0.0112$ (iKD TgAP2IX-5 +auxin compared to iKD TgAP2IX-5 −auxin); mean ± s.d. ($n = 3$ independent experiments). **c** Bar graph representing chromo1: nucleus ratio using the parental and iKD TgAP2IX-5 strains in the absence and presence of auxin treatment for 6 h; A Student's t-test was performed, two-tailed p-values: ***$p = 0.0006$ (iKD TgAP2IX-5 +auxin compared to Parental Tir1 −auxin), ***$p = 0.0007$ (iKD TgAP2IX-5 +auxin compared to Parental Tir1 +auxin), **$p = 0.0022$ (iKD TgAP2IX-5 +auxin compared to iKD TgAP2IX-5 −auxin); mean ± s.d. ($n = 3$ independent experiments).

affected, the number of parasites with an elongated plastid was recorded in the presence and absence of TgAP2IX-5. We observed a significantly high number of elongated plastid in the absence of TgAP2IX-5 (Fig. 4e), suggesting a critical role for TgAP2IX-5 after elongation and before plastid division. This short timeframe corresponds to TgAP2IX-5 protein cell-cycle-dependent expression. As expected, the mitochondria division (the last step during the cell cycle, before cytokinesis) is also affected in the iKD TgAP2IX-5 mutant (Supplementary Fig. 5c and 5d), confirming that the blockage during the cell cycle precedes mitochondria division.

**TgAP2IX-5 impacts the expression of cell-cycle-regulated genes**. Since TgAP2IX-5 is a potential transcription factor, we examined the changes in the transcriptome after depletion of the TgAP2IX-5 protein using RNA-seq. Total RNA was purified from tachyzoites of the iKD TgAP2IX-5 strain grown with or without auxin for 6 h (three biological triplicates). Data analysis using Deseq2 allowed us to identify significant changes in the iKD TgAP2IX-5 transcriptome with an adjusted p-value cutoff of 0.05 and a minimum fold change of 2 (Fig. 5a). We identified more than 600 transcripts that were downregulated and around 300 transcripts (Supplementary Data 1) that were upregulated in the iKD TgAP2IX-5 mutant when treated with auxin (Fig. 5a). We examined the cell cycle expression of the downregulated genes and represented their expression using a heatmap (Fig. 5b). We discovered that most of them showed an expression peak during the late S and M phase with a few of them exhibiting a peak during the cytokinesis phase (Fig. 5b) while the upregulated genes showed mostly peaks of expression that are quite heterogenous along the cycle with low expression peaks corresponding to the

S/M and cytokinesis phase. Additionally, an expression peak during the G1 phase was present (Supplementary Fig. 6a).

In the list of downregulated transcripts, we were struck by the number of annotated genes corresponding to proteins targeted to the IMC and to the apical complex, both structures that are the first to appear when the daughters bud within the mother cell. To better assess the potential localization of the proteins that correspond to downregulated transcripts, we used a HyperLopit proteomic dataset that predicts with high confidence the localization of proteins in the parasite[24]. We showed that a high proportion of the downregulated transcripts present in the dataset (331 genes) encode proteins predicted to localize to the IMC or to the apical complex (a total of 30% of the downregulated transcripts present in the HyperLopit dataset; Supplementary Fig. 6b). Moreover, these two localizations are overrepresented (16% and 14%, respectively) when compared to the whole proteome localization (3% for each; 2487 proteins). Based on this dataset, the downregulated genes represent 64% of the IMC proteome (52/81) and more than 70% (45/63) of the predicted apical complex proteome. These results suggest that TgAP2IX-5 may act mainly as an activator of genes whose expression shows a cell-cycle-regulated profile with a peak at the late S and M phases. These genes represent a high proportion of the IMC and apical complex proteome. We also examined the upregulated gene list using the same dataset[24]. While no upregulated genes are predicted to encode proteins that localize to the IMC or the apical compartment, a majority of the upregulated proteins may localize to the nucleus, ER and rhoptry (Supplementary Fig. 6c).

By RNA-seq, we were able to identify that TgAP2IX-5 either activates directly or indirectly the expression of genes. In order to identify what promoters are directly targeted by TgAP2IX-5, we

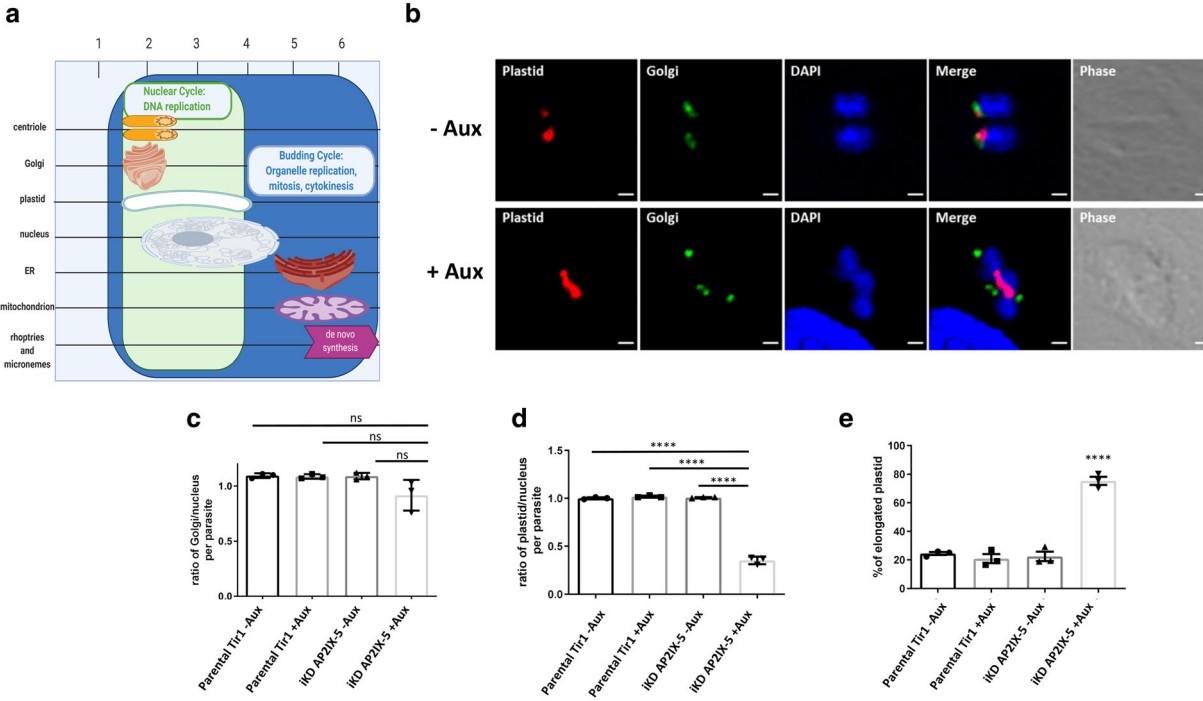

**Fig. 4 Organelle replication in iKD TgAP2IX-5 throughout the tachyzoite asexual cell cycle. a** Schematic representation of the chronological order of organellar division throughout a normal *Toxoplasma gondii* cell cycle. Nuclear cycle is indicated in green and budding cycle is indicated in blue. Timeframe of each organelle division is represented by length of representative organelle. **b** Confocal microscopy of iKD TgAP2IX-5 parasite with labelled plastid (red) and Golgi (green) in the presence and absence of overnight auxin treatment. The lower panel clearly represents the elongated plastid phenotype. DAPI was used to stain the nucleus. Scale bar is indicated at the lower right side of each image. **c** Bar graph representing the ratio of Golgi: nucleus using the parental and iKD TgAP2IX-5 strains in the absence and presence of overnight auxin treatment. A Student's *t*-test was performed, two-tailes *p*-values: $p = 0.0948$ (iKD TgAP2IX-5 +auxin compared to Parental Tir1 −auxin), $p = 0.1039$ (iKD TgAP2IX-5 +auxin compared to Parental Tir1 +auxin), $p = 0.1030$ (iKD TgAP2IX-5 +auxin compared to iKD TgAP2IX-5 −auxin); mean ± s.d. ($n = 3$ independent experiments). **d** Bar graph representing the ratio of plastid to nucleus using the parental and iKD TgAP2IX-5 strains in the absence and presence of overnight auxin treatment. A Student's *t*-test was performed, two-tailed *p*-values: ****$P < 0.0001$; mean ± s.d. ($n = 3$ independent experiments). **e** Bar graph representing the percentage of parental and iKD TgAP2IX-5 parasites with an elongated plastid in the absence and presence of overnight auxin treatment. A Student's *t*-test was performed, two-tailed *p*-values: ****$P < 0.0001$; mean ± s.d. ($n = 3$ independent experiments).

carried out a ChIP-seq analysis. Biological duplicates were produced and processed for sequencing (Supplementary Fig. 7a). The MACS2 software was used to identify the significant peaks ($p$ value < 0.05) that were in intergenic regions close to an annotated gene. This analysis revealed that TgAP2IX-5 directly binds to 696 gene promoters among which key genes are involved in parasite formation (Fig. 5c and Supplementary Fig. 7a–d) such as genes encoding proteins targeted to the IMC such as TgIMC1 and TgIMC4 (Fig. 5ci), TgGAPM3 (Fig. 5cii), and TgIMC29 (Fig. 5ciii). Interestingly, we also identified that TgAP2IX-5 was able to bind to the promoters of other ApiAP2 encoding genes (Fig. 5civ and Supplementary Fig. 7c–d). We examined the cell cycle expression of the genes whose promoter was targeted by TgAP2IX-5 and found that a majority of these genes were showing an expression peak during the S and M phase, although a cluster of genes showed a strong expression during C and early G1 phase (Supplementary Fig. 7e). These data confirm the ability of TgAP2IX-5 to act as a bona-fide transcription factor by directly binding to promoters of genes encoding essential proteins for the establishment of the daughter cells.

Since RNA-seq does not enable the identification of the genes that are solely directly controlled by TgAP2IX-5, we overlapped the RNA-seq and ChIP-seq dataset. For that, we identified within the differentially regulated genes list whether they were down-regulated or upregulated (as initially identified from RNA-seq), and compared this list with the genes identified to be targeted by TgAP2IX-5 from ChIP-seq analysis using the MACS software.

Overall, 117 genes were recorded from the overlap of RNA-seq and ChIP-seq data representing a 17% overlap (Fig. 5d). A closer study of the overlap between RNA-seq and ChIP-seq genes identified that 10% of upregulated genes are directly targeted by TgAP2IX-5 whereas 14% of downregulated genes are directly targeted by TgAP2IX-5. We examined the cell cycle regulation of the downregulated transcripts directly controlled by TgAP2IX-5 and found that these genes exhibit a cell-cycle-regulated expression peak during the late S and early M phases (Fig. 5e). These results suggest that TgAP2IX-5 directly controls genes expressed during the S phase and beginning of the M phase. Examples of genes that were found to be directly activated by TgAP2IX-5 included a number of IMC proteins such as TgIMC1, TgIMC4, TgIMC3, TgIMC29, and TgGAPM3 (Supplementary Data 2 and Fig. 5c). Since TgAP2IX-5 was shown to directly activate genes that are mainly involved in daughter parasite formation, TgIMC29 was myc-tagged within the iKD TgAP2IX-5 strain and we observed a significant decrease in the TgIMC29 expression in the absence of TgAP2IX-5 as confirmed by Western blot (Supplementary Fig. 8a).

Considering the difference between the RNA-seq and ChIP-seq dataset, we searched the RNA-seq dataset for potential regulators that might be directly regulated by TgAP2IX-5 and therefore may in turn perturb the expression of genes not directly targeted by TgAP2IX-5. Interestingly, eight ApiAP2 TFs were downregulated (TgAP2III-1, TgAP2III-2, TgAP2IV-4, TgAP2VIIa-1, TgAP2X-11, TgAP2XI-4, TgAP2XII-9, and TgAP2XII-2) and four upregulated

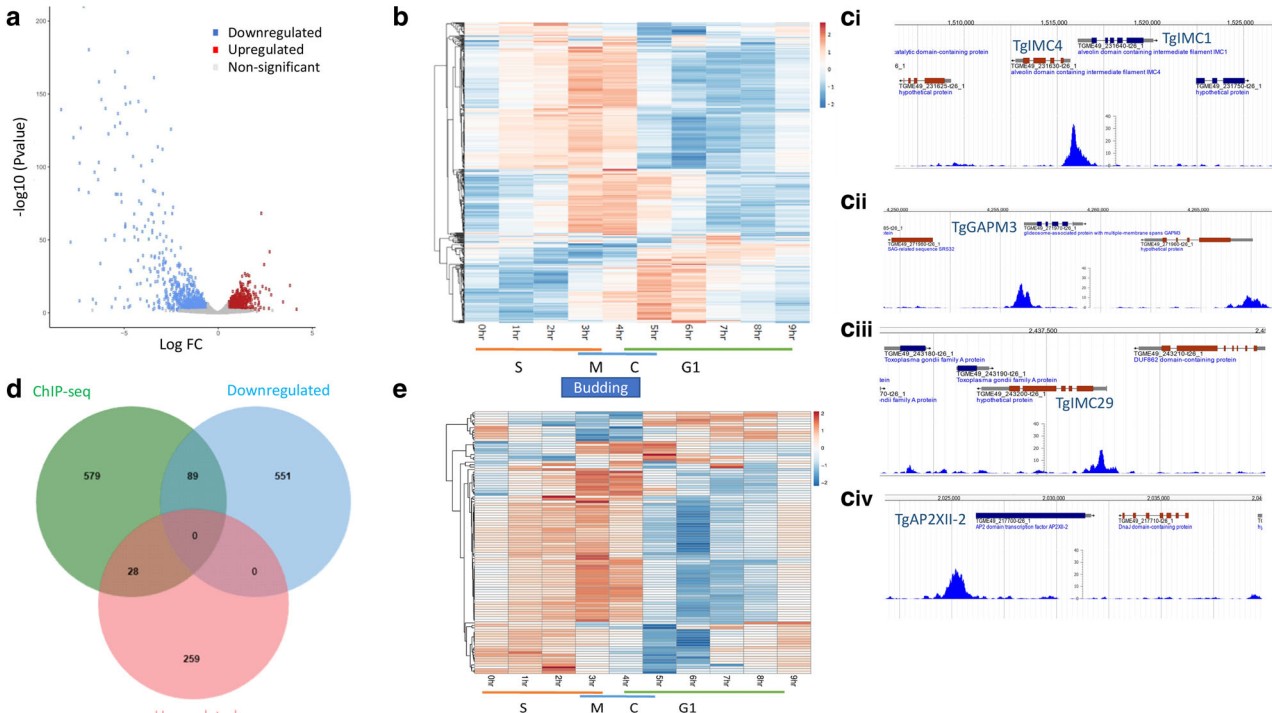

**Fig. 5 TgAP2IX-5 controls the expression of key genes involved in daughter parasite formation. a** Volcano plot of differentially expressed genes analyzed from RNA-sequencing of TgAP2IX-5 parasites treated with auxin for 6 h. Downregulated genes are represented in blue, upregulated genes are represented in red. Statistically nonsignificant genes are represented in gray. The differential expression analysis (DE) was based on three independent biological experiments. **b** Heatmap showing the cell cycle expression of all individual transcripts that are downregulated in the iKD AP2IX-5 strain in the presence of 6 h of auxin treatment. The cell cycle phases are represented at the bottom as well as the timing when budding occurs. **c** ChIP-seq data representing the direct targeting of TgAP2IX-5 to the promoters of TgIMC4 (i), TgIMC1 (i), TgGAPM3 (ii), TgIMC29 (iii), and TgAP2XII-2 (iv) genes. MACS2 generated tracks are represented together with the annotated genes (top). **d** Venn diagram of overlapping downregulated genes, upregulated genes from RNA-seq, and identified promoters of genes directly interacting with TgAP2IX-5. DEseq2 and MACS2 software were used to analyze RNA-seq and ChIP-seq data, respectively. **e** Heatmap of 89 downregulated genes directly activated by TgAP2IX-5. The cell cycle phases are represented at the bottom.

(AP2IX-5, AP2IX-1, AP2VI-3, and AP2X-9) following the depletion of TgAP2IX-5 as measured by RNA-seq (Supplementary Data 1). Among these genes, five (TgAP2IV-4, TgAP2III-2, TgAP2XII-9, TgAP2X-9, and TgAP2XII-2) had their promoters directly bound by TgAP2IX-5 (Supplementary Fig. 8b, underlined). Interestingly, much like other genes directly regulated by TgAP2IX-5 (Fig. 5e), these genes showed an expression peak during the late S phase (Supplementary Fig. 8b) with the exception of TgAP2X-9 which peaks in early S phase and whose expression is likely directly repressed by TgAP2IX-5 (upregulated in absence of TgAP2IX-5 and its promoter bound by TgAP2IX-5). TgAP2IV-4, a known repressor of developmentally regulated genes, is expressed during the S/M phase[17]. Since TgAP2IV-4 is involved in differentiation, we examined the expression of the upregulated genes expression profile during the parasite lifecycle (Supplementary Fig. 8c). Interestingly, TgAP2IX-5 depletion induced the expression of transcripts that are preferentially expressed in bradyzoites and also during the sexual stages that occur in the definitive host (Supplementary Fig. 8c). Genes preferentially expressed in bradyzoite include MAG1 a known cyst matrix protein[25]. Interestingly, a cluster of genes upregulated in absence of TgAP2IX-5 is strongly expressed in the early days of sexual development where division by endopolygeny occurs (Supplementary Fig. 8c, EES1 and EES2). These data suggest that TgAP2IX-5 directly controls other TFs during the S phase that may in turn activate the late S and M expression program but also coordinate developmental choices (such as differentiation into bradyzoite).

Surprisingly, we found that TgAP2IX-5 was enriched at its own promoter (Supplementary Fig. 7d, boxed) and the TgAP2IX-5 transcript was found to be upregulated in the presence of auxin based on RNA-seq. These data indicate that TgAP2IX-5 may directly regulate its own transcript expression suggesting that TgAP2IX-5 expression may be under the direct regulation of a possible negative feedback loop.

**Complementation demonstrates TgAP2IX-5 is responsible for the phenotypes observed.** We generated a complemented strain (iKDc TgAP2IX-5) by inserting a myc-tagged version of the TgAP2IX-5 gene (under the control of its own promoter) into an exogenous locus (*uprt*; Supplementary Fig. 9a). The expression and localization of the exogenous TgAP2IX-5-myc in the complemented strain was verified by immunofluorescence (Supplementary Fig. 9b) and the percentage of positively labelled parasite with myc-tag was compared to the expression of the endogeneous HA-tagged copy. A similar number of parasites (30% of the asynchronous parasite population) were shown to express the myc-tagged copy in the complemented strain rather than the parental iKD TgAP2IX-5 strain (Supplementary Fig. 9c–d). In order to determine whether the iKD TgAP2IX-5 strain phenotype can be complemented by the ectopic expression of TgAP2IX-5, the number of nuclei per parasite was recorded in the iKDc TgAP2IX-5 strain in the absence and presence of auxin. We observed that the maximum number of nuclei per parasite did not exceed two nuclei per parasite (Supplementary Fig. 9e). These

results demonstrated that the multinucleated phenotype of iKD TgAP2IX-5 is due to the absence of the TgAP2IX-5 protein. Similarly, Golgi to nucleus and plastid to nucleus ratios were recorded. Calculated ratios of around 1:1 were recorded and we therefore inferred that each parasite contains one Golgi as well as one plastid (Supplementary Fig. 9f–g). These results demonstrate that the TgAP2IX-5 protein was indeed responsible for the phenotypes observed.

**TgAP2IX-5 regulates cell cycle pattern flexibility from endodyogeny to endopolygeny.** The timing of daughter cell formation is key to define the cell division pattern employed by the parasite at any given time of its lifecycle. Since we established that TgAP2IX-5 is the master regulator controlling the production of daughter cells during endodyogeny, we reasoned that controlling the expression of TgAP2IX-5 may be sufficient to switch from one division pattern to another. *T. gondii* undergoes endopolygeny, during its asexual reproduction in the definitive host, where multiple nuclei are formed before a final phase of internal budding. We created a strain of the iKD TgAP2IX-5 mutant expressing a marker of the IMC (TgIMC3-mCherry) to be able to follow the daughter cell formation using live imaging. We treated

this strain with auxin to deplete the TgAP2IX-5 protein and obtain parasites that presented around four nuclei per parasite (overnight treatment with auxin, Supplementary Fig. 10a). Using this treatment and after auxin washout, TgAP2IX-5 re-expression was apparent after 3 h (Supplementary Fig. 10b, c). By time-lapse microscopy, we were able to visualize the fate of these parasites after washing auxin out from cell culture media and inducing the re-expression of the TgAP2IX-5 protein. At $T_0$ (parasites grown for 16 h with auxin and 3 h without auxin) we observed enlarged multinucleated parasites. At $T_1$ (43 min after $T_0$), we started to observe multiple daughter cells emerging from within the initial multinucleated mother parasite labeled with TgIMC3. The emergence of parasites continued throughout $T_2$ (1.5 h after $T_0$) and $T_3$ (1.7 h after $T_0$). At $T_4$ (3.9 h after $T_0$), we observed individual parasites each separately labelled with TgIMC3-mCherry that had completely emerged from the mother parasite (Fig. 6a). After removing auxin from the media, the multinucleated parasites where daughter cell budding was inhibited by the absence of TgAP2IX-5, continued their division in a similar way to endopolygeny. Multiple daughter parasites were observed to internally bud within the cytoplasm of the mother cell leading to the release of multiple daughters from an initial multinucleated parasite (Supplementary Movie 1). This forced endopolygeny was due to

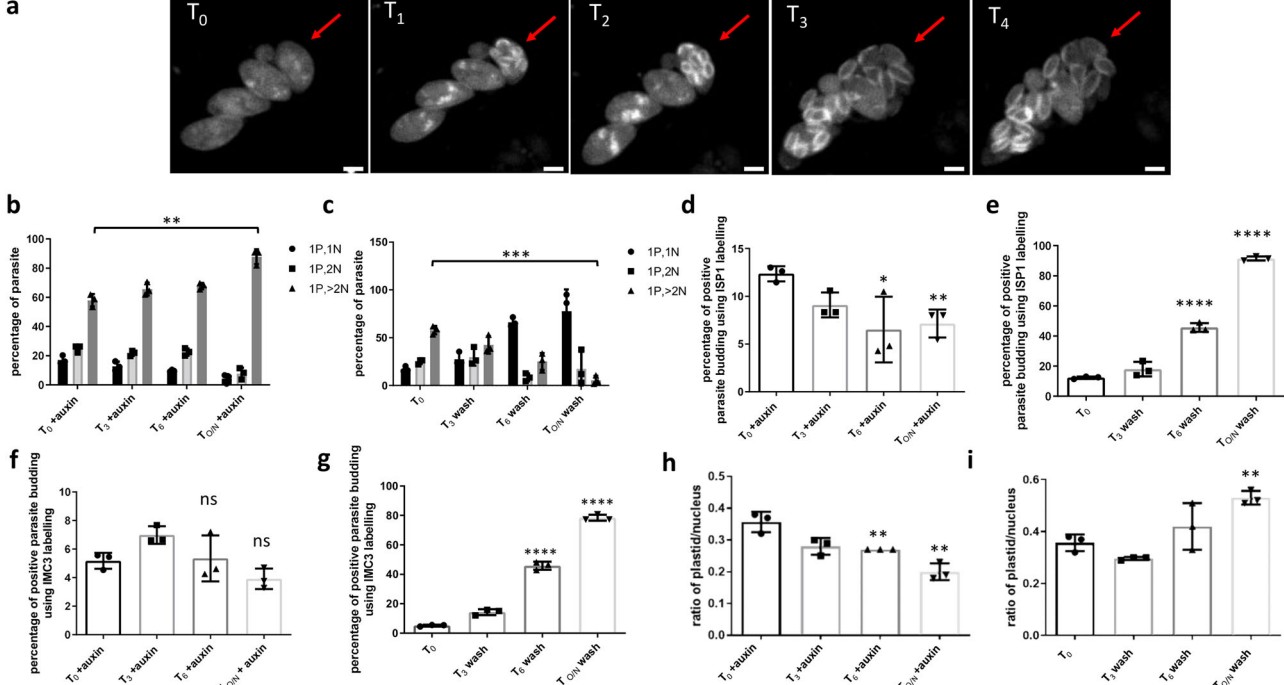

**Fig. 6 iKD TgAP2IX-5 parasites treated with auxin resume cell division and daughter parasite formation mimicking endopolyogeny. a** Video-microscopy images of iKD TgAP2IX-5 parasites at different timepoints after auxin washout. The budding vacuole and the emergence of parasites is indicated with a red arrow. The IMC of the parasite is labelled with IMC3-mCherry. The scale bar is indicated at the bottom right of each panel. **b** Bar graph representing nucleus per parasite ratio. Timepoint $T_0 = 0$ min is equal to start of auxin wash. Two-tailed *p*-values: **$p = 0.0015$; mean ± s.d. ($n = 3$ independent experiments). **c** Bar graph representing iKD TgAP2IX-5 nucleus per parasite counts during different timepoints of auxin washout treatments; 3 h, 6 h, and overnight (O/N) washout. Two-tailed *p*-values: **$p = 0.0001$; mean ± s.d. ($n = 3$ independent experiments). **d** Bar graph representing parasite budding in the iKD TgAP2IX-5 strain during overnight auxin treatment using ISP1 labelling. Two-tailed *p*-values: *$p = 0.0457$, **$p = 0.0057$; mean ± s.d. ($n = 3$ independent experiments). **e** Bar graph representing parasite budding in the iKD TgAP2IX-5 strain using TgISP1 labelling during different timepoints of auxin washout treatments; 3 h, 6 h, and overnight (O/N) washout. Two-tailed *p*-values: ****$p < 0.0001$; mean ± s.d. ($n = 3$ independent experiments). **f** Bar graph representing parasite budding in the iKD TgAP2IX-5 strain using TgIMC3 labelling during overnight auxin treatment. Two-tailed *p*-values: ns: $p > 0.05$; mean ± s.d. ($n = 3$ independent experiments). **g** Bar graph representing parasite budding in the iKD TgAP2IX-5 strain using TgIMC3 labelling during different timepoints of auxin washout treatments; 3 h, 6 h, and overnight (O/N) washout. Two-tailed *p*-values: ****$p < 0.0001$; mean ± s.d. ($n = 3$ independent experiments). **h** Bar graph representing the ratio of plastid to nucleus in the iKD TgAP2IX-5 strain during overnight auxin treatment. Two-tailed *p*-values: **$p = 0.0029$ ($T_{O/N}$ + auxin compared to $T_0$), **$p = 0.0095$ ($T_6$ + auxin compared to $T_0$); mean ± s.d. ($n = 3$ independent experiments). (i) Bar graph representing the ratio of plastid to nucleus in the iKD TgAP2IX-5 strain during different timepoints of auxin washout treatments; 3 h, 6 h, and overnight (O/N) washout. Two-tailed *p*-values: **$p = 0.0020$; mean ± s.d. ($n = 3$ independent experiments).

the re-expression of TgAP2IX-5 in these parasites. These data indicate that expression of TgAP2IX-5 controls the initiation of the budding process and that parasites initially depleted from TgAP2IX-5 were still division competent.

To quantify the ability of the parasite to restart division after an initial depletion of TgAP2IX-5, we designed a protocol producing parasites with around four nuclei and then observed the effect of the re-expression of TgAP2IX-5 on the number of nuclei per parasite. For that, we counted the number of nuclei per parasite by immunofluorescence at different timepoints after auxin washout. While this number increased with longer auxin treatment (Fig. 6b), we observed that the number of nuclei per parasite decreases as the duration of auxin washout increases (Fig. 6c) indicating that the parasites formed by the "forced" endopolygeny are competent for the next cycles of division. However, we noticed a small percentage of parasites with multiple nuclei after overnight auxin washout, indicating that some parasites did not recover from the original depletion of the TgAP2IX-5 protein. In order to study the effect of auxin removal on daughter parasite formation, we recorded the ability of the initial budding capability of the parasite by labelling TgISP1 and TgIMC3 during auxin washout and compared it with the parasite's budding capability when continuing the auxin treatment. We observed a significant increase in the daughter parasite's budding ability when removing auxin that increased as the duration of auxin washout increased (Fig. 6d, g, Supplementary Fig. 10d). To assess if the parasites were able to divide and segregate the apicoplast after depletion and re-expression of TgAP2IX-5, we monitored the fate of the plastid during auxin washout by recording the ratio of plastid to nucleus. While we observed a steady decrease of the number of plastid per nucleus in presence of continuing auxin treatment (Fig. 6h), we recorded an increase in the ratio of plastid to nucleus per parasite as the duration of auxin washout increased (Fig. 6i) demonstrating that the plastid was competent for replication. These results demonstrate that the timing of TgAP2IX-5 expression is the sole determinant of the creation of daughter cells in the parasite. It may therefore determine the cell division pattern flexibility from endodyogeny to endopolygeny observed in this study. In order to determine whether the parasites generated after re-expression of TgAP2IX-5 (auxin washout) and produced by forced endopolygeny remain viable, we carried out plaque assays. For that, the parasites were left to grow in the presence of auxin for 16 h, 24 h, and 48 h before auxin washout and removal. They were then grown without auxin for several days. Plaques were visible after 16 h and 24 h auxin treatment and subsequent washout. Residual plaques were visible after 48 h auxin treatment (Supplementary Fig. 11). This indicates that the parasites emerging from a division cycle by forced endopolygeny were viable.

## Discussion

TgAP2IX-5 depletion was found to completely halt the formation of daughter cells (budding cycle) while nuclear division seemed unaffected. When auxin was added at the time of parasite invasion, daughter cell budding was stopped in the first cell cycle, indicating a direct relationship between budding and the expression of TgAP2IX-5. This led to the accumulation of multinucleated parasites without any signs of daughter cell formation. To our knowledge, such a stark phenotype was observed before in a collection of temperature sensitive cell cycle mutants[9], over-expressing a dominant negative version of TgRAb11b[26], and when mutating the specific fibers (SFA2 and 3) that connect the centrosome (outer core) to the forming daughter cells[5]. However, this is the first report demonstrating the importance of

transcriptional control in the timing and assembly of daughter cell formation.

After TgAP2IX-5 depletion, the cell cycle stops at a specific timepoint after plastid elongation and before its segregation. This timepoint corresponds to the onset of daughter cell formation. Surprisingly, the nuclear cycle proceeds while the apicoplast remains elongated and is not segregated. This underlines the absence of a checkpoint to ensure proper apicoplast segregation as was observed for other mutants[27]. A similar effect on plastid division was observed upon depletion of MORN1[28] or in a TgDrpA mutant[29] although they were able to form daughter cells.

Flexibility between the nuclear and budding cycle is thought to be controlled in this parasite by the dual core centrosome with the inner core controlling the nuclear cycle and the outer core controlling the budding cycle[10]. A detailed study of the effect of TgAP2IX-5 on the subcellular structures of the parasite revealed that nuclear and bipartite centrosome division remain mainly unaffected by TgAP2IX-5 depletion. Given the lack of daughter parasite formation observed after TgAP2IX-5 depletion, it is surprising that the outer core centrosome (as represented by TgCentrin1) is duplicated but remains inactivated by the parasite's kinases, such as TgMAPK-L1[10]. In addition, TgAP2IX-5 may participate in the regulation of the expression of TgFBOX1 because it is found present at its promoter. However, TgFBOX1 transcript (which has a transient expression during the cell cycle) is not present in the RNA-seq dataset. TgFBOX1 localizes early at the daughter cell bud and may organize the daughter cell scaffold[30]. Depletion of FBOX1 does not lead to such a dramatic effect on budding like those observed in the iKD TgAP2IX-5 strain. Other components of the outer core, such as the SFA fibers that were shown to be involved in the emergence of daughter cells[5], but were present at the centrosome after TgAP2IX-5 depletion. Indeed, SFA fiber expression was also unchanged in the mutant as measured by RNA-seq. This indicates that the outer core centrosome functionality is probably intact in the mutant but centrosome activation and maturation is lacking to proceed with the budding cycle. Therefore, it is most likely that depletion of TgAP2IX-5 hinders normal centrosomal activity despite its successful division. The experiments inducing re-expression of TgAP2IX-5 after its depletion further illustrate the undamaged function of the centrosome in absence of the TF, since daughter cell formation re-started after several cycles of unproductive budding. TgAP2IX-5 is therefore the determinant factor for centrosome activation and the control of the budding cycle.

The phenotype observed after TgAP2IX-5 depletion illustrates the independence of the nuclear and budding cycles in this parasite. Using inducible degradation of the TgAP2IX-5 protein, we have established that the tachyzoites are able to divide by endopolygeny in vitro. These experiments also underline the ability of the centrosome to duplicate and maintain its functionality in absence of daughter cell formation. This is a key aspect of the parasite centrosome biology that allows to accumulate nuclei and then activate the budding cycle as seen for division patterns such as endopolygeny or schizogony. Although tachyzoites divide by endodyogeny, they retained the ability to divide by endopolygeny. This simple mechanism of control, which allows multiple nuclear cycles to happen before cytokinesis is also dependent on TgAP2IX-5. The presence of the TgAP2IX-5 transcript during asexual division in the intermediate host in the tachyzoite and bradyzoite stage (endodyogeny) or asexual division in the definitive host[31] (endopolygeny) may indicate that the same mechanisms are shared for both division patterns, although global expression profiles are profoundly different. Regulation of the timing of expression of TgAP2IX-5 may be sufficient to switch from one division pattern to another. Alternatively, the absence of TgAP2IX-5 may be a signal that promotes

endopolygeny. In this line, it is interesting to note that the artificial depletion of TgAP2IX-5 induced the expression of a set of specific transcripts that are normally expressed in the first days of the sexual cycle when endopolygeny occurs. Strikingly, two ApiAP2 (AP2IX-1 and AP2VI-3) TFs upregulated after depletion of TgAP2IX-5 are normally expressed in the sexual stages, indicating that in absence of TgAP2IX-5 part of a sexual specific expression program may be promoted. Interestingly, PF3D7_0613800, the AP2 domain containing protein that is homologous to AP2IX-5 is expressed at the end of the red blood stage (schizont stage) when parasites are budding after multiple round of DNA synthesis. The molecular mechanisms that count the rounds of DNA synthesis and provide the proper timing for parasite budding may be linked to the proteins that activate the expression of TgAP2IX-5.

We have established that TgAP2IX-5 controls the timing of activation of the centrosome and therefore creation of daughter cells. However, the molecular mechanisms leading to centrosome activation remain unknown. This key event permits the production of daughter cells at the right timing irrespective of the division pattern used. We reasoned that TgAP2IX-5 may directly control the expression of the proteins in charge of centrosome activation. Although expression of the SFA fibers or other known centrosome markers were unaffected in the mutant, we noticed that the expression of a key centrosomal protein, TgCep530, was directly under the control of TgAP2IX-5 (downregulated in RNA-seq and its promoter bound in ChIP-seq). This protein was shown to be targeted to a centrosomal region situated between the inner and outer core[32]. It is essential for the coordination of the nuclear and budding cycle although budding seemed to occur[32]. However, its depletion leads to the accumulation of outer core centrosome[32], a phenotype that is not observed with TgAP2IX-5 expression. This protein may therefore have other functions in the maturation or activation of the outer core centrosome that were not identified previously. Another possibility leading to lack of centrosome activation is a plausible downstream effect of nontranscriptional origin, targeting either a kinase or phosphatase with an important role in the centrosome activation cascade. Key serine/threonine kinases such as MAPK-like protein kinases have been identified within *T. gondii* and have been shown to

ensure proper formation of new daughter parasites[9,10]. We found that expression of kinases and phosphatases were downregulated after TgAP2IX-5 depletion, but in our ChIP-seq analysis, TgAP2IX-5 was not found to bind their promoters. One obvious candidate, that is directly regulated by TgAP2IX-5, was the TgCDC48AP protein but it was shown to be involved in apicoplast protein import[33]. TgAP2IX-5 directly activates the expression of several proteins of unidentified function in *T. gondii*; these hypothetical proteins may serve as direct centrosome activators and the characterization of their function may lead to the discovery of new centrosome activators.

We have also established that TgAP2IX-5 directly activates key components of the daughter cell scaffold. In particular, it regulates the expression of components to be targeted early to the forming daughter buds, such as TgISP1[34], TgIMC1, TgIMC3, TgIMC4, and TgIMC10[35]. Apical cap component, such as TgAC2 and TgAC7 may also be loaded early on the forming buds. Similarly, TgGAPM3 expression is directly dependent on TgAP2IX-5 and its invalidation provokes IMC collapse[36]. However, TgIMC15, a protein known to be loaded early on the centrosome before reaching the forming buds is not affected by TgAP2IX-5 depletion. TgIMC15 may have a limited role in daughter cell formation or centrosome activation as suggested by the mild phenotype exerted by its mutant[37]. This suggests that TgAP2IX-5 may activate the centrosome but also directly regulates the expression of key components of the daughter cell scaffold that will be needed in a short timeframe to proceed with daughter cell formation (Fig. 7).

We observed that RNA-seq and ChIP-seq datasets only partially overlapped. We reasoned that indirect effects on gene expression may be identified using RNA-seq even at a short timing after TgAP2IX-5 depletion (6 h). The cell cycle expression profile of the genes that are directly controlled by this TF and downregulated in its absence showed that regulated transcripts are expressed during the S phase before the budding occurs. Among these genes, we identified four other ApiAP2 transcription factors (TgAP2III-2, TgAP2IV-4, TgAP2XII-2, and TgAP2-XII-9) that were directly controlled by TgAP2IX-5. These TFs may be responsible for the expression of the transcripts that were identified as downregulated in RNA-seq and that peak during the

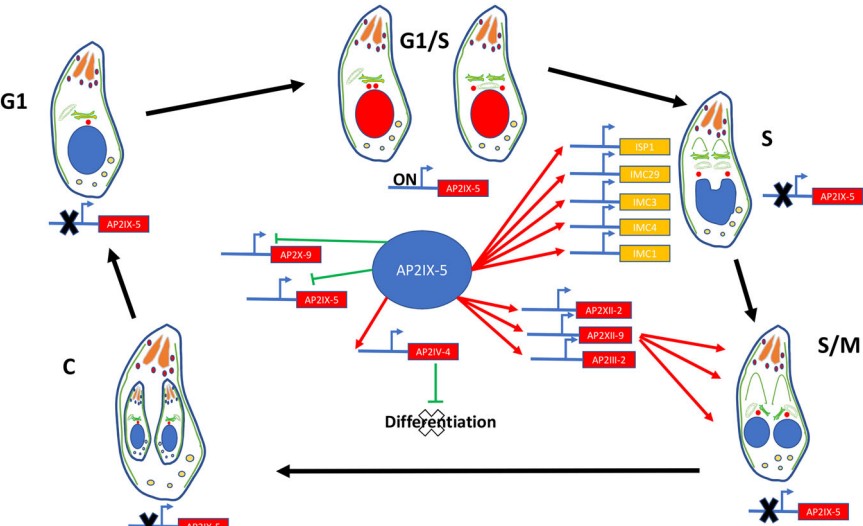

**Fig. 7 Schematic representation of the effect of TgAP2IX-5 depletion on daughter cell formation according to the different phases of the cell cycle.** Absence of TgAP2IX-5 leads to the blockage of the cell cycle within the G1/S phase. This is associated with the direct targeting of key inner membrane complex proteins such as TgISP1, TgIMC1, TgIMC29, TgIMC3, and TgIMC4 as well as other key TFs (TgAP2XII-2, TgAP2XII-9, TgAP2III-2, TgAP2IV-4, TgAP2X-9). TgAP2IV-4 is a known repressor of differentiation. TgAP2X-9 may be repressed by TgAP2IX-5. The TgAP2IX-5 transcript may also be regulated by a negative feedback loop.

late S and M phase. Although no data are available for TgAP2III-2, TgAP2XII-2, and TgAP2XII-9, TgAP2IV-4 was shown to be a repressor of the bradyzoite differentiation expression program[17]. Interestingly, depletion of TgAP2IX-5 leads to the upregulation of transcript preferentially expressed in bradyzoites, such as MAG1, indicating that the absence of TgAP2IX-5 promotes the differentiation pathway. The control of the expression of a repressor of differentiation by a TF that also controls the continuation of cell cycle may provide the missing link between cell cycle and differentiation. It has been shown that the cell cycle is linked to bradyzoite differentiation[38]. In particular, the developmental switch toward the latent bradyzoite is made during S phase and/or mitosis[38]. It seems convenient for the parasite to link the choice of continuing the tachyzoite cell cycle (by starting the budding cycle) with the expression of a repressor of differentiation. Therefore, expression of TgAP2IX-5 may serve as a molecular checkpoint for the choice between proliferation or differentiation. TgAP2IX-5 may act as a limiting factor for this developmental choice at each round of the cell cycle (Fig. 7).

We also observed that TgAP2IX-5 may act as a repressor since it was found at the promoter of genes whose transcript was upregulated after depletion of this protein. It was striking to see that the TgAP2IX-5 transcript was overexpressed after TgAP2IX-5 depletion. We also found that TgAP2IX-5 was bound to its own promoter indicating that a direct negative feedback loop may be present to limit the expression of TgAP2IX-5. This might explain the short timeframe when TgAP2IX-5 is expressed during the cell cycle since its expression might be self-limiting. Another ApiAP2 TF expression (TgAP2X-9) may be directly repressed by TgAP2IX-5. TgAP2X-9 transcript peaks at the G1/S boundary, just before the expression peak of TgAP2IX-5. Therefore, TgAP2IX-5 may repressed the expression of the TFs that control the cell cycle expression program preceding the budding phase. In summary, TgAP2IX-5 serves as a platform for the repression of the G1/S expression program (through TgAP2X-9 repression), the promotion of the S/M expression program (through the activation of TgAP2III-2, TgAP2XII-2, and TgAP2XII-9) and limiting differentiation (through AP2IV-4 activation).

In other eukaryotes, TFs associate with each other and form complexes to either activate or repress gene expression[39]. ApiAP2s are known to associate with each other[20] and differential association may impact their activity.

TgAP2IX-5 is therefore an essential regulator during the T. gondii tachyzoite cell cycle. TgAP2IX-5 controls the activation of the outer core centrosome to induce the budding cycle. It also controls the expression of key proteins for formation of the daughter cell scaffold, ensuring that the proper continuation of the cell cycle is achieved. TgAP2IX-5 is also a master regulator that controls the expression of the TFs that are no longer needed and those that will be further needed for the completion of the budding cycle. It also acts as a limiting factor that ensures that asexual proliferation continues by promoting the inhibition of the differentiation pathway at each round of the cell cycle. TgAP2IX-5 is therefore a master regulator that controls cell cycle and developmental pathways. However, several questions remain to be answered. For example, what are the proteins whose expression are controlled by TgAP2IX-5 and that are responsible of centrosome activation? Moreover, the molecular mechanisms that allow the same TF to act as an activator or a repressor are still unknown and will be the subject of further studies.

## Methods

**Parasite culture, transfection, and purification**. *Toxoplasma gondii* tachyzoites of the RH Δ*ku80* Tir1 strain were propagated in vitro in human foreskin fibroblasts (HFF) using Dulbeccos's modified Eagles medium supplemented with 10% fetal calf serum (FCS), 2 mM glutamine, and 1% penicillin–streptomycin. The RH

Δ*ku80* Tir1 strain is a rapidly growing strain, which contains a knock-out of the *ku80* gene increasing the likelihood of homologous recombination to occur and the transfection to succeed. This strain also expresses the Tir1 protein that allows the inducible degradation of proteins that are tagged with a specific recognition sequence after addition of auxin in the culture media. The AID system was initially described in ref. [40]. Tachyzoites were grown in ventilated tissue culture flasks at 37 °C and 5% $CO_2$ in a HERAcell VLOS 160i $CO_2$ incubator (Thermo Scientific). Transgenes were introduced by electroporation using a BTX Harvard apparatus elctroporator (ECM 630) into tachyzoites of *T. gondii* strains and stable transformants were selected by culture in the presence of either 25 µg/ml mycophenolic acid (MPA) and 50 µg/ml xanthine, pyrimethamine (2 µM) or FUDR (5 µg/ml). Clonal lines were obtained by limiting dilution. Prior to total RNA, genomic DNA or protein purification, intracellular parasites were purified by sequential syringe passage with 17-gauge and 26-gauge needles (Terumo AGANI needles) and filtration through a 3-µm polycarbonate membrane filter (Whatman)

**Generation of transgenic *T. gondii* strains**. The iKD TgAP2IX-5 line was generated using RH Δ*ku80* Tir1 strain[40] using a Cas9 plasmid targeting the 3' end of the gene after the stop codon and a PCR product representing the HA-AID cassette flanked by homology regions. To produce Myc-tagged TgSFA2 (TGME49_205670) or TgIMC29 (TGME49_243200) parasite lines, a DNA fragment representing the tag and the selection cassette was amplified using the pLIC-Myc-DHFR plasmid as a template. The PCR product (10 µg) and a pSAG1::Cas9-U6 targeting the 3' UTR of the respective genes were transfected in the iKD TgAP2IX-5 strain. The sequences of all primers used in this study are listed in Supplementary Table 1. In order to generate the iKD complementation line, 60 µg of a plasmid containing 3-kb upstream the predicted ATG of the TgAP2IX-5 gene and the full-length c-myc-tagged TgAP2IX-5 gene flanked by 2 kb homology fragments for the *uprt* gene was cotransfected with the pSAG1::Cas9-U6::sgUPRT plasmid in the iKD TgAP2IX-5 strain to ensure insertion into the UPRT locus. The parasites were then selected using 5 µM 5-fluoro-2'-deoxyruridine (FUDR). To obtain the iKD TgAP2IX-5 strain expressing TgIMC3-mcherry, the iKD TgAP2IX-5 strain was transfected by a plasmid expressing TgIMC3-mcherry under the *tubulin* promoter and selected using pyrimethamine.

**Growth assays**. For the growth assay, $8 \times 10^4$ of parental and iKD TgAP2IX-5 parasites at a multiplicity of infection of around 2 (calculated host cell to parasite ratio = 1.8) per well of a 24-well plate were inoculated on HFF cells grown on coverslips for 24 h in normal media or media treated with 0.5 mM of auxin (IAA/indoleacetic acid). Auxin was added in order to induce the degradation of the TgAP2IX-5 protein. The coverslips were then fixated using 4% paraformaldehyde (PFA). The parasite nuclei were stained with anti-TgEno2 and anti-TgIMC1 antibodies, respectively, and the number of parasites per vacuole was counted. A total of 100 vacuoles were counted for each replicate. Three independent experiments were performed. For the plaque assay, two hundred of parental and iKD TgAP2IX-5 parasites were inoculated on HFF cells grown in a 6-well plate for seven days in normal media and media treated with 0.5 mM of auxin (IAA/indoleacetic acid). Parasites were fixated and then stained using Crystal Violet.

**Organelle labelling**. Parental and iKD TgAP2IX-5 strains were inoculated on HFF cells grown on coverslips in a 24-well plate for 24 h. HFF cells were infected at a parasite to host-cell ratio of 1.8. This was then followed by the addition of auxin for 3 h, 6 h, and 12 h prior to fixation with 4% PFA. Auxin was added in order to induce the depletion of the TgAP2IX-5 protein. The nucleus was labelled with anti-TgEno2 and the number of nuclei per parasite was counted. For Inner Membrane Complex labelling, parental and iKD TgAP2IX-5 strains were left to grow for 24 h on HFF grown on coverslips for 24 h. This was then followed by the addition of auxin for 6 h. Intracellular parasites were labelled using anti-TgISP1 and anti-TgIMC1. Components of the centrosome were labelled after leaving parental and iKD TgAP2IX-5 strains to grow overnight followed by auxin treatment for 6 h and TgCentrin1 and TgChromo1 labelling. Golgi, plastid, and mitochondrion were labelled after leaving parental and iKD TgAP2IX-5 strains to grow overnight in auxin with anti-TgCT Sort, anti-TgACP and anti-TgTom40, respectively.

**Immunofluorescence assays (IFA)**. All IFAs were carried out using the similar following protocol. Intracellular parasites were fixed using 4% PFA for 30 min. The coverslips were incubated with primary antibodies and then secondary antibodies coupled to Alexa Fluor-488 or to Alexa-Fluor-594. Primary antibodies used for IFAs include: anti-TgEno2, anti-TgISP1[34], anti-TgIMC1 (a gift from Pr. Ward, U. Vermont), anti-TgCentrin1 (a gift from Pr. Gubbels, Boston College), anti-TgChromo1[22], anti-TgSortilin (Golgi), anti-TgACP (Plastid), anti-TgTom40[41] (mitochondrion), anti-HA (Sigma-Aldrich) and anti-myc (abcam) antibodies were used at the following dilutions: 1:1000, 1:500, 1:500, 1:500, 1:500, 1:500, 1:500, 1:500, 1:500, 1:200, respectively. Confocal imaging was performed with a ZEISS LSM880 Confocal Microscope or Apotome Microscope at 63 magnification. All images were processed using Carl Zeiss ZEN software. Quantification of immunofluorescence assays was carried out manually by counting the concerned signal corresponding to the organelle by visual observation. Signal corresponding to 100 parasites was counted for each replicate.

**Electron microscopy**. HFF monolayer was inoculated with iKD TgAP2IX-5 strain. iKD strain was grown in either normal media or media treated with auxin for 24 h. Transmission electron microscopy samples were fixed with 1% glutaraldehyde in 0.1 M sodium cacodylate pH 6.8 buffer, at 4 °C overnight. They were postfixed with 1% osmium tetroxide and 1.5% potassium ferricyanide then with 1% uranyl acetate, both in distilled water at room temperature in the dark, for 1 h. After washing, samples were dehydrated with increasing ethanol-concentration solutions. Samples were finally infiltrated with epoxy resin and cured at 60 °C for 24 h. Sections of 70–80 nm thickness deposited on formvar-coated grids were observed at 80 kV with a Hitachi H7500 TEM (Milexia, France), and images were acquired with a 1 Mpixel digital camera from AMT (Milexia, France).

**Western blotting**. Total protein extracts of $2 \times 10^6$ of parental and iKD TgAP2IX-5 parasites grown for 24 h in either normal media or media treated with auxin were resuspended in 1X SDS buffer. The protein samples were then fractionated on a 6% SDS-polyacrylamide electrophoresis gel and then transferred onto a nitrocellulose membrane (GE Healthcare) Chemiluminescent detection of bands was carried out by using Super Signal West Femto Maximum Sensitivity Substrate (Thermo Scientific).

**Live imaging**. The transgenic parasite line iKD TgAP2IX-5 expressing TgIMC3-mCherry was directly inoculated on HFF monolayers grown in an 8-well chamber. The parasites were left to grow for 7 h before adding 0.5 mM Auxin for 16 h. Auxin was washed out from the medium by changing normal cell culture media three times. Fluorescent signals were collected sequentially every 13 min, with an average of 4 lines, a zoom factor (varying between 2 and 4) and resulting in images that were 512 × 512 pixels in size, and 8 bits in resolution (256 gray levels). The images were treated with ImageJ (NIH). Z-stack acquisitions enabled to visualize the 3D localization of fluorescent signals. Movies were created at a rate of 2 frames per second. T0 = base line for washout, T1 = 43 min after auxin washout, T2 = 1.5 h after auxin washout, T3 = 1.73 h after auxin washout, T4 = 3.9 h after auxin washout.

**TgAP2IX-5 re-expression experiments**. Parasites of the iKD TgAP2IX-5 strain were inoculated on HFF cells grown on coverslips in a 24-well plate for 16 h in the presence of auxin. Auxin was washed out and parasites were left to grow in auxin-free medium for different durations of time for 3 h, 6 h, and 16 h before fixation with 4% PFA (Supplementary Fig. 12) Parasite nuclei were labelled with TgEno2 and the number of nuclei per parasite were counted. In similar experiments, the parasite's inner membrane complex (IMC) was labelled with TgISP1 and TgIMC3 and the percentage of daughter parasite formation was measured. In a third comparable experiment, plastid was labelled using anti-TgACP and the ratio of plastid to nuclei was measured. Three independent experiments were performed for each experiment.

**RNA sample preparation and extraction**. RNA samples were prepared by infecting T175 flasks containing HFF cell monolayers with iKD TgAP2IX-5 parasite for 24 h followed by 6 h auxin treatment before sample collection and addition of Trizol (Invitrogen). Control samples were left to grow in normal media. Auxin was added for 6 h since this is the duration required for T. gondii tachyzoites to complete a full cycle of asexual division. RNA was extracted as per manufacturer instruction. This was then followed by genomic DNA removal and cleaning using the RNase-free DNase I Amplification Grade Kit (Sigma). All RNA samples were assessed for quality using an Agilent 2100 Bioanalyzer. RNA samples with an integrity score greater than or equal to 8 were included in the RNA library preparation. Triplicate (biological replicate) were produced for each conditions.

**RNA library preparation**. The TruSeq Stranded mRNA Sample Preparation kit (Illumina) was used to prepare the RNA libraries according to the manufacturer's protocol. Library validation was carried out by using DNA high-sensitivity chips passed on an Agilent 2100 Bioanalyzer. Library quantification was carried out by quantitative PCR (12 K QuantStudio).

**Chromatin Immunoprecipitation and library preparation**. ChIP was performed using a protocol previously described[20]. Briefly, intracellular parasites were grown 40 h and fixed for 10 min at room temperature using 1% formaldehyde. DNA was processed by sonication using the Bioruptor device for 10 min at 4 °C with a 30 sec on/off cycle. Protein-DNA complexes were then diluted in IP Dilution Buffer (16.7 mM Tris-HCl pH8, 167 mM NaCl, 1.2 mM EDTA, 0.01% SDS, 1.1% Triton and 0.5 mM PMSF) and incubated ON with IgG Agarose beads (Amersham Biosciences). The next day, beads were washed five times with ChIP wash buffer (50 mM Tris-HCl pH8, 250 mM NaCl, 1% NP-40, 1% desoxycholic acid and 0.5 mM PMSF) and eluted twice in 75 µL of ChIP elution buffer (50 mM NaHCO₃ and 1% SDS). Duplicate immunoprecipitation and input samples (biological replicate) were produced. Purified DNA (1 ng) was used as a template for library preparation using the Nugen Ovation® Ultralow System V2 kit according to the manufacturer's instructions. Libraries were validated using a Fragment Analyzer and quantified by qPCR (ROCHE LightCycler 480).

**Sequencing, preprocessing, and dataset cleaning**. Clusters were generated on a flow cell with within a cBot using the Cluster Generation Kit (Illumina). Libraries were sequenced as 50 bp-reads on a HiSeq 2500 using the sequence by synthesis technique (Illumina). HiSeq control software and real-time analysis component were used for image analysis. Illumina's conversion software (bcl2fastq 2.17) was used for demultiplexing. Raw sequencing datasets for RNA-seq (110 M total reads) and ChIP-seq (28 M total reads) were quality-checked with FastQC v0.11.8-0. Sequencing adapters were trimmed with Cutadapt v1.18[42]. Finally, low-quality bases (LEADING:20, TRAILING:20, SLIDINGWINDOW:4:25) were trimmed and reads shorter than 30 bp were filtered out using Trimmomatic v0.38.1[43]. FastQC was used to quality-check each intermediary dataset and the final cleaned datasets (105 M cleaned RNA-seq reads, 23 M cleaned ChIP-seq reads).

**RNA-seq analysis**. Cleaned datasets (2 conditions ∗ 3 biological replicates ∗ 2 technical replicates = 12 datasets) were aligned with HiSAT2 v2.1.0[44] against the T. gondii ME49 genome from ToxoDB-39[45] (93 M aligned reads), and expression for annotated genes was quantified using htseq-count (union mode) from the HTSeq suite v0.9.1. Raw counts from technical replicates were added before performing differential expression analysis between iKD TgAP2IX-5 samples treated with or without auxin for 6 h with DESeq2 v1.22.1 using the SARTools framework v1.6.6[46] with the default parameters (fitType = "parametric", alpha = 0.05, pAdjustMethod = "BH", cooksCutoff = TRUE, independentFiltering = TRUE). P values for multiple testing were adjusted using the Benjamini–Hochberg method. Differentially expressed genes with adjusted p-value below 0.05 and log2 fold change above 2 were kept. The dataset is deposited in the GEO database under the accession number GSE150406.

**ChIP-seq analysis**. Cleaned datasets (Input & IP ∗ 2 biological replicates ∗ 2 technical replicates = 8 datasets) were aligned with Bowtie2 v2.3.4 against the T. gondii ME49 genome from ToxoDB-39[45] (14 M aligned reads). Alignments from biological and technical replicates were merged using SAMtools v1.9 and duplicates were identified with Picard MarkDuplicates v2.18.20. ChIP quality and dataset consistency were checked using the deepTools suite v3.1.3[47]. Fragment size was estimated and peaks were called with MACS2 v2.1.2[48]. Additional bigwig tracks comparing ChIP with IP were generated with MACS2 bdgcmp. The dataset is deposited in the GEO database under the accession number GSE150406.

**Statistics and reproducibility**. All data were analyzed with Graph Pad Prism software version 8 (San Diego, California, USA). Differences in the means were assessed by Student's t-test. In all cases, p values are two-sided and $P < 0.05$ was considered as significant. All experiments were repeated at least three times using biologically independent replicates. A minimum of 100 parasites was scored in each independent experiment.

**Reporting summary**. Further information on research design is available in the Nature Research Reporting Summary linked to this article.

## Data availability
RNA-seq and ChIP-seq data that support the findings of this study have been deposited in GEO database under the accession number GSE150406. Source data are provided with this paper.

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

## Acknowledgements

We wish to thank the BioImaging Center Lille for access to instruments and Lille University for access to the high-performance computing resources and the bilille cloud. This work was supported by Centre National de la Recherche Scientifique (CNRS), Institut National de la Santé et de la Recherche Médicale (INSERM), grants from the French National Research Agency (ANR) [grant number ANR-13-JSV3-0006-01 to M.G.] and the CPER CTRL Longévité (to Bilille).

## Author contributions

A.S.K.: Data collection, drafting the manuscript, data analysis, and interpretation; C.G.S.: Data collection; K.M.L.: Data collection, data analysis, and interpretation; L.H.: Data collection; T.M.: Data collection; P.P.: Data analysis and interpretation; N.B.: Data collection; H.T.: Data analysis and interpretation; G.M.: Data analysis and interpretation, E.R.: Critical revision of the article, drafting the manuscript; M.G.: Conception or design of the work, drafting the manuscript, data analysis, and interpretation.

## Competing interests

The authors declare no competing interests.
