## [Peer Review File · Nature Communications]

Reviewer comments, first round -

Reviewer #1 (Remarks to the Author):

Summary

In this manuscript it is shown that controlled TgAP2IX-5 expression is directly promoting the expression genes encoding daughter cytoskeleton while at the same time suppressing genes, which include an AP2 needed for bradyzoite differentiation (providing putative mechanism connecting known insights in connection between cell cycle and differentiation), as well as itself. The combination of RNA-seq and ChIP-seq data are powerful to differentiate direct from indirect effects. The presented findings and insights are very significant as AP2IX-5 is identified as master switch to turn on the budding cycle, and when it is turned off only the nuclear cycle progresses. Importantly, it is reversible, although some data and description are lacking to fully interpret and evaluate the extend of the findings. There are also discrepancies between data presented in Suppl Table S3, the text body and other figures that need attention. Finally, some insights on the plastid and early daughter formation were missed in interpretation and discussion.

Specific points

1. Elongated plastid in Fig 3E and results line ~435; discussion line ~665. Plastid division has been studied in quite some detail (Striepen lab papers) and completion of division is dependent on DrpA and the basal complex (MORN1). Please also refer to the lower panels of Fig 3B (either in legend or results) that this represents the elongated plastid phenotype (I assume that is why Fig 3B has 2 panels +Aux).
2. IMC1 is not present in Table S3, yet is presented in Fig 4Ci as having a AP2-IX-5 binding site. This looks like a potential bi-directional promoter driving both IMC1 and IMC4?
3. IMC29 is not in Table S3, yet is presented in Fig 4Ciii. Together with the above comment, seems like Table S3 as included is not the version used to assemble Fig 4C and write the manuscript.
4. Line 527/8. "TgAP2IV-4, TgAP2III-2, TgAP2XII-9 and TgAP2XII-2" does not match up with Table S3, as XII-9 is not in the table. Am I reading this table wrong as there are so many discrepancies, or is this set with a different FDR? I am also unable to check the data deposited on GEO as it not yet made available (till 2023 it says)....
5. Please add the hyperLopit data in the Tables S2 and S3.
6. Fig 4 all panels. The font sizes used are too small and figures very hard to decipher.
7. Figure 5. This is a complementation experiment, not a reversion experiment. If the endogenous gene expression is suppressed +Aux, there is always the extra Myc-tagged allele in the FUDS locus that prevents induction of the phenotype. As such, the conclusion that "multi-nucleated phenotype of iKD TgAP2IX-5 is reversible" is false, as the phenotype is never induced in the first place. This experiment is not providing any meaningful insights and is recommended to be removed.
8. Fig 6B-I. The methodology is not clear. For example, how can ~90% of the parasites be budding ISP1 positive after o/n washout, but not after 3 or 6 hr washout? How long was the +Aux treatment before removing the block? And is 3 hr, 6 hr, and o/n really the time elapsed since doing the Aux washout (this is it according to the text section of the paper), or is this the time of auxin treatment till the washout (if so, how long after the washout were the data collected)? Either way, it shows the phenotype is reversible, but I still have a hard time grasping 90% of parasites budding, which indicates high synchrony (and thus an exciting new tool to synchronize the division cycle??). Or does this mean they initiate budding, but never produce mature daughters, which means not truly a full reversion of the phenotype. The movie is also not conclusively showing full maturation; some mother IMC remains. A plaque assay of reversion would be insightful as well to answer that question. Overall, a small schematic of the induction and reversion times of the experiment would be very helpful here.
9. Line 675 paragraph. The bipartite centrosome model is that the outercore still duplicates but does not get 'activated' by MAPK-L1 for example. In addition, an F-Box protein residing on the centrosome was recently shown to be essential to start daughter budding by Ira Blader's lab.

Minor points

1. It is nowhere indicated how many vacuoles were counted for each rep. Please add to M&M and/or legends.

2. line 172. "anti-TgPlastid" should be "anti-TgACP" (as per info in line 180)
3. Fig 1C. Showing the whole length of the a-HA blot would be appreciated to indeed see there is only a single band
4. Fig 2D, Fig6B & Fig S3. Please add in the legend what "P" and "N" abbreviations represent. I have still not figured out what the "P" stands for as it always seems to be 1 and is not variable... (as such, does not contain meaningful information); N is nuclei as far as I can tell.
5. Fig S3C: there are three asterisks (***) over the y-axis title
6. Fig 3A. some comments on how the structures are represented based on mammalian cells but not reflecting *Toxoplasma* ultrastructure: centrioles in *Toxoplasma* are parallel, not perpendicular; mitochondrial cristae are round in *Toxoplasma*, not lamellar
7. Fig 3B, C. How long have these cells been treated with Aux?

Reviewer #2 (Remarks to the Author):

Nature Communication (NCOMMS-20-18982):

The manuscript titled "A single master regulator controls asexual cell cycle division patterns in *Toxoplasma gondii*" by Khelifa and colleagues aims to characterize the role of the transcription factor TgAP2IX-5 in the asexual cell cycle division of the parasite *Toxoplasma gondii*. Although some of the findings have been described before; it is already known that "TgAP2IX-5 protein expression is tightly controlled during the cell cycle", the work is interesting and the investigators found, among many other new findings, that (1) TgAP2IX-5 plays a role in plastid division, (2) no connection exists between a multi-nucleated phenotype of the mutant strain and the budding cycles and (3) the centrosome is able to duplicate and maintains its functionality even in the absence of daughter cell formation. The present manuscript should improve the understanding of the regulation of this fundamental reproductive process in this important parasite, which may ultimately lead to identification of therapeutic interventions to inhibit the parasite cell cycle progression.

That said, the manuscript lacks clarification, and, in some places, correction is needed to improve the readability and completeness of the text. I listed some comments/suggestions for the authors to consider while revising their manuscript.

Abstract

- Not well-articulated. The text is not well jointed and jumps from point to point with no clear flow.
- The abstract lacks a hypothesis.

Introduction:

- The introduction section is long, poorly structured and reads like a review article. There are some jargons that make it difficult to see the point made by the authors. I think the whole introduction needs to be rewritten in order to make it more focussed.
- More emphasis on the mechanism of transcriptional control by TgAP2IX-5 and other related TFs will enhance the authors' argument and the need for this study and will also serve as a launching ground for the discussion of the study findings later one.
- Line 45: "parasites of high transmissibility", I do not think this applies to apicomplexan parasites, they are not contagious organisms. Perhaps, the investigators refer to the high prevalence rates of these parasites, but definitely, no high transmissibility.
- Line 69: please use an abbreviated version of the parasite name.
- Line 79: "environment", what do you mean with environment, is it the cellular microenvironment, external environment, etc, please specify.
- Line 80: 'host' again, please be more specific about the context of mentioning the host, do you mean certain animal species, specific cell type, the role of the host in the life cycle etc.
- There is an overlap between the abstract and introduction. For example, "All apicomplexan parasites have complex life cycles exhibiting division characterized by a tightly regulated cell cycle control, resulting in the emergence of daughter parasites in possession of a single nucleus and a complete set of organelles." This sentence is mentioned in the abstract lines 24-26 and again in the introduction lines 52-54. Avoid repetitions across different sections of the manuscript.

Materials and methods

- Line 125: "Parasite tissue culture and manipulation", I suggest changing this heading to "Parasite culture, transfection and purification".
- What is the strain and genotype of the parasite used? Could you justify why this strain/genotype was used in particular and not any other types?
- There is no mention of the source of reagents, kits, or instruments used in any of the described experiments.
- In the growth assays and elsewhere, what was the reason for using auxin? This should be explained. There is no place in the methods where one can find the reason for adding auxin in the experiment. An indication is provided in the results, but I think this should be also mentioned in the methods. This is the case with most of the experiments described in this manuscript, no rationale was given in the methodology section for any of the experiments conducted.
- The description of all methods is limited, no sure this is because a space limit specified by the journal, but even so, detailed methodologies that would allow other researchers to repeat the experiments should have been provided even as supplementary materials.
- Why 6 hr in particular was chosen as the most appropriate time point at which RNA-sequencing of TgAP2IX-5 strain treated with auxin was performed? Such a short time after depletion of TgAP2IX-5 might not reveal the full spectrum of transcripts whose expression were altered.
- Since there is RNASeq data anyway, I wonder why the authors did not do more computational analysis to find out more about the function of TgAP2IX-5 and its involvement in other pathways using something like gene ontology analysis, pathway analysis, protein-protein interaction analysis. This could have identified pathway highly affected by deletion of the TF.
- Also, the description of data analysis is incomplete, for example was the p-value adjusted for multiple testing using the Benjamini-Hochberg procedure? Perhaps, the investigators did that, but this needs to be mentioned in the method.
- Line 124: "HFF cells monolayer" should be "HFF cell monolayers".
- I could not see anywhere in any experiment the Multiplicity of Infection (MOI), how many tachyzoites were used per cell. It is possible that different MOIs could have an influence on the parasite behaviour within the host cells.
- The manuscript lacks any description of the quantitative analysis of the results obtained from imaging experiments, such as when quantifying the ratio of plastid to nucleus or the ratio of Golgi to nucleus between the parental and iKD TgAP2IX-5 strains.

Results

- Figure 2A: I think the 3rd bar from the left should be labelled (- Auxin, not + Auxin). Also, the difference in the number of tachyzoites per vacuole between the KO and parental strain in the presence or absence of auxin is not huge after 24 hrs. You would expect more tachyzoites within the vacuole after 24 hrs in the case of WT strain.
- Figure 2C is missing a scale bar for the TEM images that show the difference in the nucleus in TgAP2IX-5 iKD in the presence and absence of auxin.
- Figure S1. "TgAP2IX-5 is indicated in red. TgCentrin1 is indicated in green. DAPI was used to stain the nucleus. Scale bar is indicated at the lower right side of each image." This was repeated 3 times in the same legend. Again, avoid repetitions.
- Figure S3. (A) What is the difference between the mutant and WT strain in the plaque assay. I cannot see any significant differences in the disruption/integrity of the cell monolayers between the two strains based on the images provided. How were the differences quantified? This needs to be described. Also, in (C) there are some asterisks on the label of Y axis, please remove.
- Figure S6. Not sure why there are two rows for +Aux in panel A, explain in the legend please.

Discussion and conclusion:

- Line 692-693: "Although tachyzoites divide by endodyogeny, they retained this ability." Please explain retained their ability for what?
- Line 763: "licensing factor", this term is not suitable, please use a more relevant terminology to describe the effect of this TF on proliferation or differentiation. Perhaps, you could use something like "limiting factor".
- Line 772: "TF" should be "TFs". Also, please clarifies what is meant by "associate in complex"
- The conclusion is a reiteration of the main study findings, without any suggestion for the future

outlook and the next steps, where do the investigators go from here?

Reviewer #3 (Remarks to the Author):

In this manuscript Khelifa et al. present data on a transcriptional factor of the ApiAP2 family from *Toxoplasma gondii* and its role in cell cycle progression. Specifically, they show that TgAP2IX-5 is a cell cycle-regulated transcriptional factor that controls the activation of the budding-specific cell cycle expression program. In absence of TgAP2IX-5, the mutant parasites complete mitosis and karyokinesis but are unable to initiate budding, generating cells with multiple nuclei. The authors nicely demonstrate through reversible destabilization of the TgAP2IX-5 protein that its expression is sufficient for activation of the budding cycle in these multinucleated parasites, suggesting that TgAP2IX-5 might control the switch from endopolygeny to endodyogeny division patterns. Because Apicomplexa have unique modes of division, studying division is an important topic in the field. The experimental approaches used in this study and the analysis are well justified. The logic is clearly laid out for the reader to follow. Although the results might not be generalized to all apicomplexans, they are original and provide new clues about the peculiar modes of division of *Toxoplasma*, that will likely be influential within the field. However, the manuscript suffers from several approximations both in the text and in the figures that do not allow full appreciation of this original study. Moreover, some analyses are incomplete and leave several open questions.

Major comments:

1. It is overstated to say "Single » master regulator in the title. Transition between division modes involves likely several layers of regulation and factors, i.e. transcriptional factors, kinases, histone modifiers etc... «transcriptional regulator » may be also specified in the title.
2. There is not enough analysis and discussion about the genes found to be up-regulated upon TgAP2IX-5 depletion, which represent an important subset of genes and that may be as important in the endodyogeny/endopolygeny switch. More details should be provided about, for instance, the predicted location of the proteins, the overlap with ChIP-seq analysis, the presence of others AP2 factors or others transcriptional factors. It would be particularly important to know if some of these up-regulated genes are highly expressed in the cat schizont stages (that divide by endopolygeny): that would strengthen the potential role of this protein in controlling the switching between endopolygeny/endodyogeny.
3. There is no comments about the conservation of this protein among others apicomplexan parasites. If it is only present in parasites that divide by endopolygeny and endodyogeny, that would support a role in controlling the switching between endopolygeny/endodyogeny. Similarly, while transcriptomics datasets are available, the authors did not comment on the expression of AP2IX-5 in cat stages (endopolygeny) and in the chronic phase (endodyogeny).
4. The authors show down-regulation of API2IV-4 (known to inhibit bradyzoite gene expression, Radke et al 2018) in absence of AP2IX-5 and that AP2IX-5 binds to API2IV-4 promoter. This might suggest a role in controlling differentiation into the dormant bradyzoite stage. Is there any enrichment in bradyzoites genes in absence of AP2IX-5? Are bradyzoites markers expressed after depletion of AP2IX-5?
5. The reversible inhibition of budding initiation by auxin removal is spectacular but deserves some more characterization. Do the parasites generated after budding reinitiation (resulting from this artificial 'endopolygeny') remain viable and fully competent for the successive steps of the lytic cycle? Do they re-invade and form lysis plaque? If not, are the invasion organelles in place?
6. Not all parasites fully recover their ability to make daughter cells again. Is this related to the level of re-expression of AP2IX-5 that may vary between parasites? There is no IFA or WB showing correct re-expression of AP2IX-5 after auxin removal.
7. What is the rationale for investigating the location of IMC29 in the mutant since the authors

nicely demonstrate that there is no IMC anymore in absence of AP2IX-5? Instead, it would have been more useful to confirm by WB the decrease in protein expression suggested by the ChiP-seq data.

8. Inheritance of the mitochondrial network is particular difficult to outline by IFA. How was this quantified (Fig. S6C)? Can a representative picture be shown?

9. Finally, data presentation has to be improved to fully support the conclusions drawn by the authors.

-Most of IFA pictures are too small and not cropped to focus only on parasites (Fig. 1A, Fig. 3B, Fig. 4C, Fig. 6A). Add a zoom of on the IMC in the EM picture (-auxin). Some panels duplicate each other (merge Dapi, merge with DIC) and are unnecessary.

-The color code of pie charts makes it difficult to understand the data (Fig. S7).

-DIC is missing in some image series where it would be particularly important, because the number of nucleus does not necessarily reflect the number of cells in this mutant (Fig. 1A, Fig. S1A-C, Fig. S11). Alternatively, the authors may delineate the parasite shape and indicate nascent parasite buds for the non-specialist readership.

-The figures are not always well chosen and sometimes do not support the conclusions of the paper. Fig. 2B is supposed to illustrate the absence of budding (+Auxin), but the control parasites '-auxin' are not in a budding state. In Fig. 2B, the '+auxin' vacuole has two parasites, while the graph and text say that the mean of number of parasites per vacuole is 1 after depletion of AP2IX-5. Similarly, in S4B, there are 4 parasites. I guess, these pictures were from experiments in which the auxin has been added several hours post-infection. To avoid any confusion, this need to be specified in the text. Fig. 3C: the picture is not appropriate as on the bottom there are 5 DAPI-stained nuclei for two Golgi.

-Some important data are in supplementary material (centrosome and centromere division Fig. S5 ; heatmap of upregulated genes S7A), while others which are in my opinion unnecessary are part of the main figures (schematic of the auxin degron system, Fig. 1B ; Fig4C, ChIP-seq data representing the direct binding of TgAP2IX-5 on the promoters of IMC genes). The cell cycle profile of TgAP2IX-5 might be shown in Fig. 1.

-Also, the final schematic is slightly confusing as it stands, it needs to be reworked to be more explicit.

Minor comments :

1. Most of figures contain oversized parts next to parts which are too tiny. Size of the text is not homogenous and some text is barely readable on many figures. Scales on IFA pictures are too small.

2. Line 46. Toxoplasma is a single species, remove 'spp'.

3. Lines 49-50 « Although, apicomplexan parasites usually present a sexual cycle within the definitive host, ... » Awkward wording: all Apicomplexa have a sexual stage, and by definition a definitive host is where sexual reproduction happens, so why say "usually"?

4. It is not clearly stated in the introduction that the asexual reproduction of Toxoplasma in its definitive host is by endopolygeny (l. 74).

5. M&M line 155, correct the sentence: The parasite nuclei and the IMC were stained with anti-TgEno2 and anti-TgIMC1 antibodies, respectively, and the number of parasites per vacuole was counted.

6. M&M RNA sample preparation : It is not clear if the samples were treated for 6 hours or 24 hours?

7. Lines 453-456. Not an 'inverse expression profile'. Peaks of expression are quite heterogenous along the cycle, but low expression points, however, do correspond to the S/M and cytokinesis phase.

8. Figure 1A (As in many figures) the IFAs are too small, this not possible to appreciate whether or not the centrosomes are divided but remain in close proximity of each other (Centrin1). Add DIC or outline the parasites.
9. Line 311, TgAP2IX-5 gene in italic
10. Line 336, write the exact mean of parasite per vacuole, which is more than 1.
11. Line 371 ; we measured the number of parasites per nuclei. It should be « the number of nuclei per parasite ».
12. Fig. 2D. Propidium iodide and flow cytometry analysis would give clues as to which extent nuclear replication continues.
13. The authors say that the recorded ratios of centrosome to nucleus and centromere to nucleus are close to 1 in both the presence and absence of auxin, however the Student's t-tests in the figure S5 and S6 show statistically lower numbers for both centromeres and centrosome OC. This have to be clearly mentioned in the text.
14. Fig. S7A. To be more accurate, the authors should precise that although the majority of up-regulated genes are not expressed in S/M, a cluster of genes showed a strong expression during this phase.
15. The panels on figure S6 are not labelled.
16. Text in figure 5 is too small.
17. Line 543, the text say « ...was verified by immunofluorescence » : the IFA pictures should be shown.
18. Line 767 : 300 genes which are up-regulated, is not a 'small number' as currently stated in the results section. (see comment 2).

Reviewer #1:

Specific Points:

1. Elongated plastid in Fig 3E and results line ~435; discussion line ~665. Plastid division has been studied in quite some detail (Striepen lab papers) and completion of division is dependent on DrpA and the basal complex (MORN1).
Elongated plastid in Fig 3E: we added additional text concerning that a similar phenotype of elongated plastid and defect in plastid segregation was observed in previous studies. Line 614-616 now reads: “A similar effect on plastid division was observed upon depletion of MORN1³⁹ or in a TgDrpA mutant⁴⁰ although they were able to form daughter cells.”.
Please also refer to the lower panels of Fig 3B (either in legend or results) that this represents the elongated plastid phenotype (I assume that is why Fig 3B has 2 panels +Aux).
The lower panel of Figure 3B is referred to in the figure 4B legend and line 912 now reads: “The lower panel clearly represents the elongated plastid phenotype.”.
2. IMC1 is not present in Table S3, yet is presented in Fig 4Ci as having a AP2-IX-5 binding site. This looks like a potential bi-directional promoter driving both IMC1 and IMC4?
IMC1 is now present in Table S3 and was not included since the peak was attributed to the IMC4 gene. Indeed, this is a potential bi-directional promoter driving both IMC1 and IMC4.
3. IMC29 is not in Table S3, yet is presented in Fig 4Ciii. Together with the above comment, seems like Table S3 as included is not the version used to assemble Fig 4C and write the manuscript.
IMC29 is now present in Table S3 and was previously included but primarily annotated as a hypothetical protein.
4. Line 527/8. “TgAP2IV-4, TgAP2III-2, TgAP2XII-9 and TgAP2XII-2” does not match up with Table S3, as XII-9 is not in the table. Am I reading this table wrong as there are so many discrepancies, or is this set with a different FDR? I am also unable to check the data deposited on GEO as it not yet made available (till 2023 it says)....
**Line 494-495. “TgAP2IV-4, TgAP2III-2, TgAPXII-9, TgAP2X-9 and TgAP2XII-2” is now matching with Table S2 and S3. The new Table S3 has also additional information as per the reviewer’s comments. A reviewer token was initially available to consult the data deposited on GEO. Please use the information below to access the data:
ChIP-seq data have been deposited on GEO with the accession number GSE150406.
Go to <https://www.ncbi.nlm.nih.gov/geo/query/acc.cgi?acc=GSE150406>
Enter token krmryeigvkhxfap into the box**
5. Please add the hyperLopit data in the Tables S2 and S3.
HyperLopit data is now included in Tables S2 and S3.
6. Fig 4 all panels. The font sizes used are too small and figures very hard to decipher.

Figure 4 all panels. The font sizes have been enlarged and figures are now easier to decipher.

7. Figure 5. This is a complementation experiment, not a reversion experiment. If the endogenous gene expression is suppressed +Aux, there is always the extra Myc-tagged allele in the FUDS locus that prevents induction of the phenotype. As such, the conclusion that “multi-nucleated phenotype of iKD TgAP2IX-5 is reversible” is false, as the phenotype is never induced in the first place. This experiment is not providing any meaningful insights and is recommended to be removed.

We agree with the reviewer and changed the terminology used in lines 526 and 527 that now read: “iKD TgAP2IX-5 strain phenotype can be complemented by the ectopic expression of TgAP2IX-5.”. We think these experiments are important to show that the sole gene responsible for the phenotype observed is TgAP2IX-5 and therefore decided to keep this data but moved it as a supplementary figure (Figure S9).

8. Fig 6B-I. The methodology is not clear. For example, how can ~90% of the parasites be budding ISP1 positive after o/n washout, but not after 3 or 6 hr washout? Regarding the methodology used to count positive budding: A multinucleated parasite was considered to be positively budding if we detected the presence of daughter parasite ISP1 caps within the vacuole which was initially multinucleated. Each count for parasite budding represents positive budding of this “vacuole”.

How long was the +Aux treatment before removing the block? And is 3 hr, 6 hr, and o/n really the time elapsed since doing the Aux washout (this is it according to the text section of the paper), or is this the time of auxin treatment till the washout (if so, how long after the washout were the data collected)?

The auxin treatment was for 24 hours before washout. Washout timepoints are as indicated within the text. 3 hours, 6 hours, and O/N is the real time that was elapsed since doing the auxin washout. The data was collected after 3 hours, 6 hours, and O/N since auxin washout and this is the time of fixation of the coverslips.

Either way, it shows the phenotype is reversible, but I still have a hard time grasping 90% of parasites budding, which indicates high synchrony (and thus an exciting new tool to synchronize the division cycle??).

We show that 90% of the vacuoles showed signs of budding but they do not reach synchrony.

Or does this mean they initiate budding, but never produce mature daughters, which means not truly a full reversion of the phenotype. The movie is also not conclusively showing full maturation; some mother IMC remains. A plaque assay of reversion would be insightful as well to answer that question. Overall, a small schematic of the induction and reversion times of the experiment would be very helpful here.

We observed that some of the vacuoles were not able to recover from the treatment (see figure 6). The re-expression of TgAP2IX-5 allows for the initiation of budding as well as for the development of mature parasites. Following the reviewer’s suggestion we performed plaque assay and the results are now included within the paper: Lines 590-597

now read: “In order to determine whether the parasites generated after re-expression of TgAP2IX-5 (auxin wash-out) and produced by forced endopolygony remain viable, we carried out plaque assays. For that, the parasites were left to grow in the presence of auxin for 16 hours, 24 hours, and 48 hours before auxin washout and removal. They were then grown without auxin for several days. Plaques were visible after 16hr and 24 hr auxin treatment and subsequent washout. Residual plaques were visible after 48 hours auxin treatment (Figure S11). This indicates that the parasites emerging from a division cycle by forced endopolygony were viable.” A schematic representation is now included as Figure S10A depicting the time points for auxin treatment and washout.

9. Line 675 paragraph. The bipartite centrosome model is that the outercore still duplicates but does not get ‘activated’ by MAPK-L1 for example. In addition, an F-Box protein residing on the centrosome was recently shown to be essential to start daughter budding by Ira Blader’s lab

We changed the text according to the reviewer’s comment see lines 622-629 now read: “it is surprising that the outer core centrosome (as represented by TgCentrin1) is duplicated but remains inactivated by the parasite’s kinases, such as TgMAPK-L1⁹. In addition, TgAP2IX-5 may participate in the regulation of the expression of TgFBOX1 because it is found present at its promoter. However, TgFBOX1 transcript (which has a transient expression during the cell cycle) is not present in the RNA-seq dataset. TgFBOX1 localizes early at the daughter cell bud and may organize the daughter cell scaffold⁴¹. Depletion of FBOX1 does not lead to such a dramatic effect on budding like those observed in the iKD TgAP2IX-5 strain.”.

Reviewer #1 Minor Points:

1. It is nowhere indicated how many vacuoles were counted for each rep. Please add to M&M and/or legends.
The number of vacuoles that were counted for each replicate is 100 and has been added to the Materials and Methods section as well as the legend of Figure 2A. Therefore, lines 169-170 now read: “A total of 100 vacuoles were counted for each replicate.”. Lines 882-883 now reads: “A total of 100 vacuoles were counted for each replicate.”
2. line 172. “anti-TgPlastid” should be “anti-TgACP” (as per info in line 180)
Line 186 now reads: “...strains to grow overnight in auxin with anti-TgCT Sort, anti-TgACP and anti-TgTom40”.
3. Fig 1C. Showing the whole length of the a-HA blot would be appreciated to indeed see there is only a single band
The full length picture for each Western-Blot is provided as supplementary material.
4. Fig 2D, Fig6B & Fig S3. Please add in the legend what “P” and “N” abbreviations represent. I have still not figured out what the “P” stands for as it always seems to be 1 and is not variable.... (as such, does not contain meaningful information); N is nuclei as far as I can tell.

Figure 2D, Figure 6B and Figure S3. The legends now include what “P” and “N” abbreviations stand for. Lines 896-897 now read: “P stands for parasites and N stands for nuclei”. Line 928 now reads: “P stands for parasites and N stands for nuclei”. Line 936 and 939 now reads: “P stands for parasites and N stands for nuclei.”.

5. Fig S3C: there are three asterisks (***) over the y-axis title

Figure S3C: The three asterisks have now been removed from the top of the y-axis.

6. Fig 3A. some comments on how the structures are represented based on mammalian cells but not reflecting Toxoplasma ultrastructure: centrioles in Toxoplasma are parallel, not perpendicular; mitochondrial cristae are round in Toxoplasma, not lamellar

Figure 3A: The figure has now been updated to reflect the Toxoplasma ultrastructure, The centrioles are now parallel and the mitochondrial cristae are now round. This is the updated figure, the centrioles are now parallel instead of perpendicular and the mitochondria now has circular cristae instead of lamellar ones.

7. Fig 3B, C. How long have these cells been treated with Aux?

Figure 3B,C have been changed to Figures 4B,C: These cells have been treated with auxin overnight. This information can be found within the legend. See lines 912 and 914.

Reviewer #2:

Abstract

The abstract was reworked and now includes a hypothesis. Lines 31-32 now read: “We hypothesized that a transcriptional regulator may determine the timing of the budding expression program.”.

Introduction

- The introduction section is long, poorly structured and reads like a review article. There are some jargons that make it difficult to see the point made by the authors. I think the whole introduction needs to be rewritten in order to make it more focused.
For the sake of shortening the introduction, some sections have been removed. Line 88-92 have been omitted as well as lines 95-96 and lines 99-103.
- More emphasis on the mechanism of transcriptional control by TgAP2IX-5 and other related TFs will enhance the authors’ argument and the need for this study and will also serve as a launching ground for the discussion of the study findings later one.
We detailed some of the knowledge of the ApiAP2 TFs as suggested by the reviewer.
- Line 45: “parasites of high transmissibility”, I do not think this applies to apicomplexan parasites, they are not contagious organisms. Perhaps, the investigators refer to the high prevalence rates of these parasites, but definitely, no high transmissibility.
Line 46-47 now reads: “Apicomplexa is a phylum consisting of unicellular, obligate, intracellular protozoan parasites which includes various human pathogen species”. “of high transmissibility” has been removed.
- Line 69: please use an abbreviated version of the parasite name.
Line 73: *Toxoplasma gondii* has been removed and replaced by its abbreviation *T. gondii*.
- Line 79: “environment”, what do you mean with environment, is it the cellular microenvironment, external environment, etc, please specify.
Line 84: environment here refers to the cellular microenvironment. Line 84 now reads: “are able to employ several division patterns depending on the cellular microenvironment.”
- Line 80: ‘host’ again, please be more specific about the context of mentioning the host, do you mean certain animal species, specific cell type, the role of the host in the life cycle etc.
Line 84: host here refers to the intermediate host of the parasite’s life cycle which can be any warm-blooded animal species. Line 84 now reads: “and intermediate host of the parasite’s life cycle.”.
- There is an overlap between the abstract and introduction. For example, “All apicomplexan parasites have complex life cycles exhibiting division characterized by a tightly regulated cell

cycle control, resulting in the emergence of daughter parasites in possession of a single nucleus and a complete set of organelles.” This sentence is mentioned in the abstract lines 24-26 and again in the introduction lines 52-54. Avoid repetitions across different sections of the manuscript.

Overlap between abstract lines 25-27 and introduction lines 54-56. The introduction has been modified and now lines 54-56 read: “All Apicomplexa possess complex life cycles consisting of parasite propagation controlled by the tight regulation of the cell cycle and result in the formation of new daughter cells containing one nucleus and a complete set of organelles.”.

Materials and Methods

- Line 125: “Parasite tissue culture and manipulation”, I suggest changing this heading to “Parasite culture, transfection and purification”.

Line 130 now reads: “Parasite culture, transfection and purification.”

- • What is the strain and genotype of the parasite used? Could you justify why this strain/genotype was used in particular and not any other types?

The strain of *T. gondii* parasite used is an RH strain deleted for the *ku80* gene abbreviated as $\Delta ku80$. This strain also expresses the Tir1 protein. The RH $\Delta ku80$ Tir1 strain was used in particular because it is easy to carry out a transfection using this strain since the *ku80* gene is knocked out and thus this increases the chance of homologous recombination to occur and the transfection to function properly. In addition to this, this strain expresses Tir1 that allows the auxin-degradation system to function. The Tir1 protein functions through a specific recognition sequence leading to the subsequent degradation of the targeted protein. Therefore, lines 145-149 now read: “The RH $\Delta ku80$ Tir1 strain is a rapidly growing strain which contains a knock-out of the *ku80* gene increasing the likelihood of homologous recombination to occur and the transfection to succeed. This strain also expresses the Tir1 protein that allows the inducible degradation of proteins that are tagged with a specific recognition sequence after addition of auxin in the culture media. The AID system was initially described in ²²”.

- There is no mention of the source of reagents, kits, or instruments used in any of the described experiments.

More details have been added in regards to the source of reagents, kits, or instruments used in the described experiments. As such, lines 134-137 now read: “Tachyzoites were grown in ventilated tissue culture flasks at 37°C and 5% CO₂ in a HERAcell VLOS 160i CO₂ incubator (Thermo Scientific). Transgenes were introduced by electroporation using a BTX Harvard apparatus electroporator (ECM 630). Lines 140-142 now read: “intracellular parasites were purified by sequential syringe passage with 17-gauge and 26-gauge needles (Terumo AGANI needles) and filtration through a 3- μ m polycarbonate membrane filter (Whatman)”. Lines 217-219 now read: “...then transferred onto a nitrocellulose membrane (GE Healthcare)”. Chemiluminescent detection of bands was carried out by using Super Signal West Femto Maximum Sensitivity Substrate (Thermo Scientific)”.

- In the growth assays and elsewhere, what was the reason for using auxin? This should be explained. There is no place in the methods where one can find the reason for adding auxin in the experiment. An indication is provided in the results, but I think this should be also mentioned in the methods. This is the case with most of the experiments described in this manuscript, no rationale was given in the methodology section for any of the experiments conducted.

It is now stated in the materials and methods section that the reason for adding auxin is to activate the AID degradome system and degrade TgAP2IX-5. Lines 166-167 now reads: “Auxin was added in order to induce the degradation of the TgAP2IX-5 protein.”. Lines 177-178 now read: “Auxin was added in order to induce the depletion of the TgAP2IX-5 protein.”.

- The description of all methods is limited, no sure this is because a space limit specified by the journal, but even so, detailed methodologies that would allow other researchers to repeat the experiments should have been provided even as supplementary materials.

More details were added to the methods section.

- Why 6 hr in particular was chosen as the most appropriate time point at which RNA-sequencing of TgAP2IX-5 strain treated with auxin was performed? Such a short time after depletion of TgAP2IX-5 might not reveal the full spectrum of transcripts whose expression were altered.

The 6 hours time-point was chosen since 6 hours is the duration required for *T. gondii* to complete a single cell cycle. Lines 244-246 now read: “Auxin was added for 6 hours since this is the duration required for *T. gondii* tachyzoites to complete a full cycle of asexual division.”.

- Since there is RNASeq data anyway, I wonder why the authors did not do more computational analysis to find out more about the function of TgAP2IX-5 and its involvement in other pathways using something like gene ontology analysis, pathway analysis, protein-protein interaction analysis. This could have identified pathway highly affected by deletion of the TF.

Additional computational analysis was not carried out such as gene ontology analysis, pathway analysis, pathway analysis, protein-protein interaction analysis since most genes that were identified to be controlled either directly or indirectly with TgAP2IX-5 are not are annotated as “hypothetical proteins” and Toxodb database lacks additional data required for this specific analysis.

- Also, the description of data analysis is incomplete, for example was the p-value adjusted for multiple testing using the Benjamini-Hochberg procedure? Perhaps, the investigators did that, but this needs to be mentioned in the method.

The p-value adjusted for multiple testing using the Benjamini-Hochberg procedure was carried out. This is now stated in the materials and methods section and lines 288-289 now reads: “P-values for multiple testing were adjusted using the Benjamini-Hochberg method.”. More details have been added to the materials and methods section for RNA-seq analysis. Lines 287-289 now read: “with the default parameters (fitType="parametric", alpha=0.05, pAdjustMethod="BH", cooksCutoff=TRUE, independentFiltering=TRUE).”

- Line 124: “HFF cells monolayer” should be “HFF cell monolayers”.

Line 240 now reads: “HFF cell monolayers”.

- I could not see anywhere in any experiment the Multiplicity of Infection (MOI), how many tachyzoites were used per cell. It is possible that different MOIs could have an influence on the parasite behaviour within the host cells.

The multiplicity of infection for the growth assay and organelle labelling experiments is 1.8. This now stated in line 164. Lines 163-164 now read: “ For the growth assay, 8×10^4 of parental and iKD TgAP2IX-5 parasites at a multiplicity of infection of 1.8 (MOI=1.80) per well of a 24 well-plate were inoculated on HFF cells.”. Line 176 now reads: “HFF cells were infected at an MOI of 1.8.”.

- The manuscript lacks any description of the quantitative analysis of the results obtained from imaging experiments, such as when quantifying the ratio of plastid to nucleus or the ratio of Golgi to nucleus between the parental and iKD TgAP2IX-5 strains.

Quantitative analysis of the results obtained from the imaging experiments was not included since it was done manually and no software was used. All counts were carried out by visual observation with 100 parasites counted for each replicate. As such lines 199-201 now read: “Quantification of immunofluorescence assays was carried out manually by counting the concerned signal corresponding to the organelle by visual observation. Signal corresponding to 100 parasites was counted for each replicate.”.

Results

- Figure 2A: I think the 3rd bar from the left should be labelled (- Auxin, not + Auxin).
Figure 2A: The third bar from the left is now labelled -Auxin instead of +Auxin.
Also, the difference in the number of tachyzoites per vacuole between the KO and parental strain in the presence or absence of auxin is not huge after 24 hrs. You would expect more tachyzoites within the vacuole after 24 hrs in the case of WT strain
The Tir1 strain used in these experiments grows more slowly than a WT RH strain, this is reflected by the relatively low number of parasite per vacuole observed.
- Figure 2C is missing a scale bar for the TEM images that show the difference in the nucleus in TgAP2IX-5 iKD in the presence and absence of auxin.
The scale bar for the TEM images has now been added to the images that show the difference between in the nucleus in TgAP2IX-5 iKD in the presence and absence of auxin.
- Figure S1. “TgAP2IX-5 is indicated in red. TgCentrin1 is indicated in green. DAPI was used to stain the nucleus. Scale bar is indicated at the lower right side of each image.” This was repeated 3 times in the same legend. Again, avoid repetitions.
We changed the text according to the reviewer’s comment.
- Figure S3. (A) What is the difference between the mutant and WT strain in the plaque assay. I cannot see any significant differences in the disruption/integrity of the cell monolayers between the two strains based on the images provided. How were the differences quantified? This needs to be described.
The plaque assay was repeated and the image now displayed in Figure S3 demonstrates a clear difference in the integrity of the cell monolayers between the two strains. The plaque assays were quantified using visual inspection since there are no plaques present in the wells corresponding to TgAP2IX-5 in the presence of auxin.
Also, in (C) there are some asterisks on the label of Y axis, please remove.
The asterisks have been removed.
- Figure S6. Not sure why there are two rows for +Aux in panel A, explain in the legend please.

There are two panels for figure S5 with + auxin just to show two different examples of the TgiKD AP2IX-5 parasite with SFA2 labelled when TgAP2IX-5 is depleted. For the sake of clarity, we removed the second panel with +auxin.

Discussion and Conclusion:

- Line 692-693: “Although tachyzoites divide by endodyogeny, they retained this ability.” Please explain retained their ability for what?
We changed the manuscript according to the reviewer’s comment see line 646-647 that now reads: “Although tachyzoites divide by endodyogeny, they retained the ability to divide by endopolygeny.”
- Line 763: “licensing factor”, this term is not suitable, please use a more relevant terminology to describe the effect of this TF on proliferation or differentiation. Perhaps, you could use something like “limiting factor”.
‘licensing factor’ has now been replaced by ‘limiting factor’. Lines 38-39 now read: “TgAP2IX-5 acts as a limiting factor ...”. Lines 722-723 now read: “TgAP2IX-5 may act as a limiting factor for this developmental choice at each round of the cell cycle (Figure 7).” Lines 747-748 now read: “It also acts as a limiting factor that ensures that asexual proliferation continues by promoting the inhibition of the differentiation pathway at each round of the cell cycle.”
- Line 772: “TF” should be “TFs”. Line 862: TF has been replaced by TFs.
Also, please clarifies what is meant by “associate in complex”
TFs form complexes with each other and the nature of the complex might change its activity toward repression or activation of transcription. Lines 739-740 now read: “...TFs associate with each other and form complexes to either activate or repress gene expression.”
- The conclusion is a reiteration of the main study findings, without any suggestion for the future outlook and the next steps, where do the investigators go from here?
The conclusion now includes a future outlook and therefore lines 750-754 read as: “However, several questions remain to be answered. For example, what are the proteins whose expression are controlled by TgAP2XI-5 and that are responsible of centrosome activation? Moreover, the molecular mechanisms that allow the same TF to act as an activator or a repressor are still unknown and will be the subject of further studies.”

Reviewer #3:

Major Comments:

1. It is overstated to say "Single » master regulator in the title. Transition between division modes involves likely several layers of regulation and factors, i.e. transcriptional factors,

kinases, histone modifiers etc... «transcriptional regulator » may be also specified in the title.

Line 2 and therefore the title now reads: “A master transcriptional regulator controls asexual cell cycle division patterns in *Toxoplasma gondii*.”

2. There is not enough analysis and discussion about the genes found to be up-regulated upon TgAP2IX-5 depletion, which represent an important subset of genes and that may be as important in the endodyogeny/endopolygeny switch. More details should be provided about, for instance, the predicted location of the proteins, the overlap with CHIP-seq analysis, the presence of other AP2 factors or other transcriptional factors. It would be particularly important to know if some of these up-regulated genes are highly expressed in the cat schizont stages (that divide by endopolygeny): that would strengthen the potential role of this protein in controlling the switching between endopolygeny/endodyogeny.

We thank the reviewer for pointing out this gene dataset that was left out of the analysis. We looked at the predicted localization of the proteins encoded by the upregulated genes and found an over-representation of genes targeted to the nucleus (Figure S6C). We also looked at the expression of the upregulated genes (Figure S9C). We found that a number of genes upregulated in the mutant are normally expressed in the sexual stages and that two ApiAP2 TFs preferentially expressed at this stage are present in the upregulated gene dataset. Moreover, we found that TgAP2IX-5 directly represses the expression of another AP2 (AP2X-9) that may be responsible for the expression program that preceded (late G1/early S) the one controlled by TgAP2IX-5. This is now indicated in figure S8B and C. Therefore, lines 492-511 now read: “Eight ApiAP2 transcription factors were downregulated and 4 upregulated following the depletion of TgAP2IX-5 as measured by RNA-seq (Figure S8B) and showed an expression peak during the S and M phase (Figure S8B). Among these genes, 5 (TgAP2IV-4, TgAP2III-2, TgAP2XII-9, TgAP2X-9 and TgAP2XII-2) had their promoters directly bound by TgAP2IX-5 (Figure S9B, underlined). Interestingly, much like other genes directly regulated by TgAP2IX-5 (Figure 4E), these genes showed an expression peak during the late S phase (Figure S9B) with the exception of TgAP2X-9 which peaks in early S phase and whose expression is likely directly repressed by TgAP2IX-5 (upregulated in absence of TgAP2IX-5 and its promoter bound by TgAP2IX-5). TgAP2IV-4, a known repressor of developmentally regulated genes, is expressed during the S/M phase¹⁸. Since TgAP2IV-4 is involved in differentiation, we examined the expression of the upregulated genes expression profile during the parasite life cycle (Figure S8C). Interestingly, TgAP2IX-5 depletion induced the expression of transcripts that are preferentially expressed in bradyzoites and also during the sexual stages that occur in the definitive host (Figure S8C). Genes preferentially expressed in bradyzoite include MAG1 a known cyst matrix protein³⁶. Interestingly, a cluster of genes upregulated in absence of TgAP2IX-5 is strongly expressed in the early days of sexual development where division by endopolygeny occurs (Figure S8C, EES1 and EES2). These data suggest that TgAP2IX-5 directly controls other TFs during the S phase that may in turn activate the late S and M expression program but also coordinate developmental choices (such as differentiation into bradyzoite).”

3. There is no comments about the conservation of this protein among others apicomplexan parasites. If it is only present in parasites that divide by endopolygony and endodyogony, that would support a role in controlling the switching between edopolygony/endodyogony. Similarly, while transcriptomics datasets are available, the authors did not comment on the expression of AP2IX-5 in cat stages (endopolygony) and in the chronic phase (endodyogony).

TgAP2IX-5 protein is a conserved protein among Apicomplexans. This conservation is mainly focused on the AP2 domain (lines 313-314). One AP2 domain containing protein is homologous in *P. falciparum* (PF3D7_0613800) and is expressed at the schizont stage. This is now stated in lines 660-662. Based on the transcriptomics datasets available, TgAP2IX-5 is expressed during the tachyzoite, bradyzoite and the sexual stages: this is now stated in lines 649-651.

4. The authors show down-regulation of API2IV-4 (known to inhibit bradyzoite gene expression, Radke et al 2018) in absence of AP2IX-5 and that AP2IX-5 binds to API2IV-4 promoter. This might suggest a role in controlling differentiation into the dormant bradyzoite stage. Is there any enrichment in bradyzoites genes in absence of AP2IX-5? Are bradyzoites markers expressed after depletion of AP2IX-5?

We looked at the expression of genes that were upregulated in presence of auxin and found that a set of gene were preferentially expressed at the bradyzoite stage, confirming the link between the depletion of AP2IX-5 and differentiation: see figure S8C. For example, MAG1, a known cyst matrix protein is present in this list (see lines 506 and 714).

5. The reversible inhibition of budding initiation by auxin removal is spectacular but deserves some more characterization. Do the parasites generated after budding reinitiation (resulting from this artificial 'endopolygony') remain viable and fully competent for the successive steps of the lytic cycle? Do they re-invade and form lysis plaque? If not, are the invasion organelles in place?

The parasites generated after budding re-initiation resulting from the forced endopolygony remain viable and fully competent for the successive steps of the lytic cycle. By carrying out a plaque assay for re-invasion, we observe a formation of lysis plaques. Lines 590-597 now read: "In order to determine whether the parasites generated after re-expression of TgAP2IX-5 (auxin wash-out) and produced by forced endopolygony remain viable, we carried out plaque assays. For that, the parasites were left to grow in the presence of auxin for 16 hours, 24 hours, and 48 hours before auxin washout and removal. They were then grown without auxin for several days. Plaques were visible after 16hr and 24hr auxin treatment and subsequent washout. Residual plaques were visible after 48hr auxin treatment (Figure S11). This indicates that the parasites emerging from a division cycle by forced endopolygony were viable."

6. Not all parasites fully recover their ability to make daughter cells again. Is this related to the level of re-expression of AP2IX-5 that may vary between parasites? There is no IFA or WB showing correct re-expression of AP2IX-5 after auxin removal.

IFAs and a Western-blot were carried out for re-expression of TgAP2IX-5 (Figure S10B-C). These experiments show that TgAP2IX-5 is re-expressed after 3 hours of auxin washout.

7. What is the rationale for investigating the location of IMC29 in the mutant since the authors nicely demonstrate that there is no IMC anymore in absence of AP2IX-5?

Instead, it would have been more useful to confirm by WB the decrease in protein expression suggested by the ChIP-seq data.

We agree with the reviewer and performed a western-blot showing that TgIMC29 is indeed downregulated in presence of auxin. The western blot has now been added to Figure S8A. Lines 487-489 now read: “We observed a significant decrease in the TgIMC29 presence at the daughter cell budding site in the absence of TgAP2IX-5 which was further confirmed by Western blot (Figure S8A).

8. Inheritance of the mitochondrial network is particular difficult to outline by IFA. How was this quantified (Fig. S6C)? Can a representative picture be shown?

This was quantified by IFA by visual inspection. Mitochondria for 100 parasites per replicate were counted. The representative image corresponding to this mitochondrial network is now included as Figure S5D

9. Finally, data presentation has to be improved to fully support the conclusions drawn by the authors.

-Most of IFA pictures are too small and not cropped to focus only on parasites (Fig. 1A, Fig. 3B, Fig. 4C, Fig. 6A). Add a zoom of on the IMC in the EM picture (-auxin). Some panels duplicate each other (merge Dapi, merge with DIC) and are unnecessary.

Figures 1A, 3B, 4C and 6A have been enlarged and cropped to focus only on the parasites. The zoom image was added to the EM picture. Duplicate image were removed (merge with dapi).

-The color code of pie charts makes it difficult to understand the data (Fig. S7).

The pie chart was changed and the name of each category was added.

-DIC is missing in some image series where it would be particularly important, because the number of nucleus does not necessarily reflect the number of cells in this mutant (Fig. 1A, Fig. S1A-C, Fig. S11). Alternatively, the authors may delineate the parasite shape and indicate nascent parasite buds for the non-specialist readership.

The DIC image has been added to the figures.

-The figures are not always well chosen and sometimes do not support the conclusions of the paper. Fig. 2B is supposed to illustrate the absence of budding (+Auxin), but the control parasites ‘-auxin’ are not in a budding state. In Fig. 2B, the ‘+auxin’ vacuole has two parasites, while the graph and text say that the mean of number of parasites per vacuole is 1 after depletion of AP2IX-5. Similarly, in S4B, there are 4 parasites. I guess, these pictures were from experiments in which the auxin has been added several hours post-infection. To avoid any confusion, this need to be specified in the text. Fig

The picture in figure 2B (-auxin) was changed. For clarity, the legends of figure 2B and S4B now include that auxin has been added several hours post-infection. Therefore, line 892 now reads: “Auxin was added 24 hours post-infection.”. Legend of figure S4B now reads: “ (B) Bar graph representing the percentage of daughter parasite formation in the absence and presence of 6 hours of auxin treatment 24 hours post-infection...”.

3C: the picture is not appropriate as on the bottom there are 5 DAPI-stained nuclei for two Golgi.

The picture has been changed to reflect corresponding number of nuclei with Golgi.

-Some important data are in supplementary material (centrosome and centromere division Fig. S5 ; heatmap of upregulated genes S7A), while others which are in my opinion unnecessary are part of the main figures (schematic of the auxin degnon system, Fig. 1B ; Fig4C, ChIP-seq data representing the direct binding of TgAP2IX-5 on the promoters of IMC genes). The cell cycle profile of TgAP2IX-5 might be shown in Fig. 1.

We decided to include the figure S5 in the main figure (now figure 3) as the reviewer suggested. As per reviewer 2 request we kept the schematic of the auxin degnon system. We think that the direct binding to IMC genes promoters is key to demonstrate that AP2IX-5 is directly regulating genes essential for IMC formation and therefore kept that information in the main figures.

-Also, the final schematic is slightly confusing as it stands, it needs to be reworked to be more explicit.

The final schematic has been reworked to be more explicit.

Minor Comments:

1. Most of figures contain oversized parts next to parts which are too tiny. Size of the text is not homogenous and some text is barely readable on many figures. Scales on IFA pictures are too small.

The size of figures and text has been changed to be homogenous. The scale bars were increased in size.

2. Line 46. *Toxoplasma* is a single species, remove 'spp'.

Line 48 now reads: "*Toxoplasma* (cause of toxoplasmosis)"

3. . Lines 49-50 « Although, apicomplexan parasites usually present a sexual cycle within the definitive host, ... » Awkward wording: all Apicomplexa have a sexual stage, and by definition a definitive host is where sexual reproduction happens, so why say "usually"?

Lines 51-52 now reads: " Although apicomplexan parasites present a sexual cycle within the definitive host..".

4. It is not clearly stated in the introduction that the asexual reproduction of *Toxoplasma* in its definitive host is by endopolygeny (l. 74).

Lines 77-79 now read: "*T. gondii* asexually divides within its definitive host through a division scheme that closely resembles schizogony and is known as endopolygeny.".

5. M&M line 155, correct the sentence: The parasite nuclei and the IMC were stained with anti-TgEzo2 and anti-TgIMC1 antibodies, respectively, and the number of parasites per vacuole was counted.

Lines 167-169 now read: "The parasite nuclei and the IMC were stained with anti-TgEzo2 and anti-TgIMC1 antibodies, respectively, and the number of parasites per vacuole was counted.".

6. M&M RNA sample preparation : It is not clear if the samples were treated for 6 hours or 24 hours?

Lines 240-246 regarding RNA sample preparation and extraction now read: "RNA samples were prepared by infecting T175 flasks containing HFF cell monolayers with iKD TgAP2IX-5 parasite for 24 hours followed by 6 hours auxin treatment before sample collection and addition of Trizol (Invitrogen). Control samples were left to grow in normal media. Auxin was added for 6 hours since this is the duration required for *T. gondii* tachyzoites to complete a full cycle of asexual division...".

7. Lines 453-456. Not an 'inverse expression profile'. Peaks of expression are quite heterogenous along the cycle, but low expression points, however, do correspond to the S/M and cytokinesis phase
 Lines 429-434 now read: “ the upregulated genes showed mostly peaks of expression that are quite heterogenous along the cycle with low expression peaks corresponding to the S/M and cytokinesis phase. Additionally, an expression peak during the G1 phase was present (Figure S6A). Interestingly, TgAP2IX-5 depletion induced the upregulation of 4 transcripts encoding ApiAP2 transcriptions factors (AP2IX-5, AP2IX-1, AP2VI-3 and AP2X-9) (Table S2).”
8. Figure 1A (As in many figures) the IFAs are too small, this not possible to appreciate whether or not the centrosomes are divided but remain in close proximity of each other (Centrin1). Add DIC or outline the parasites.
 Figure 1A has been enlarged and the DIC added.
9. Line 311, TgAP2IX-5 gene in italic
 Line 330 now reads: “an AID-HA insert at the 3' end of the *TgAP2IX-5* gene using a CRISPR/Cas9 strategy.”
10. Line 336, write the exact mean of parasite per vacuole, which is more than 1.
 Line 342 now reads: “with a calculated mean of 1.22”.
11. Line 371 ; we measured the number of parasites per nuclei. It should be « the number of nuclei per parasite ».
 Line 360 now reads: “We measured the number of nuclei per parasite”.
12. Fig. 2D. Propidium iodide and flow cytometry analysis would give clues as to which extent nuclear replication continues.
 The cytometry analysis was attempted but the results were unsatisfying. This is due to the size of the parasite that are produced after more than 12 hours auxin treatment that make them difficult to separate from cell debris and impossible to filter using 3 or 8 µm filters. The cell debris contamination gave a background that was difficult to remove from the parasite signal. Therefore, this analysis was not included in the manuscript.
13. The authors say that the recorded ratios of centrosome to nucleus and centromere to nucleus are close to 1 in both the presence and absence of auxin, however the Student's t-tests in the figure S5 and S6 show statistically lower numbers for both centromeres and centrosome OC. This have to be clearly mentioned in the text.
 Lines 391-393 now reads: “...despite that the Student's t-test carried out for statistical analysis shows a significantly lower number for both centrosome to nucleus and centromere to nucleus ratios.”
14. Fig. S7A. To be more accurate, the authors should precise that although the majority of up-regulated genes are not expressed in S/M, a cluster of genes showed a strong expression during this phase.
 Figure S6A legend now reads: “Although the majority of up-regulated genes are not expressed in S/M, a cluster of genes showed a strong expression during this phase.”
15. The panels on figure S6 are not labelled.
 Figure S5 panels are now labelled.
16. Text in figure 5 is too small.
 Text in this figure (now S9) has been enlarged.
17. Line 543, the text say « ...was verified by immunofluorescence » : the IFA pictures should be shown.

Line 522: The IFAs corresponding to the complemented iKD TgAP2IX-5 strain is now included as Figure S9 (S9B). Lines 521-522 now read: “The expression and localization of the exogenous TgAP2IX-5-myc in the complemented strain was verified by immunofluorescence (Figure S9B).

18. Line 767 : 300 genes which are up-regulated, is not a ‘small number’ as currently stated in the results section. (see comment 2).

This comment was removed see lines 726-727.

Reviewer comments, second round -

Reviewer #1 (Remarks to the Author):

Summary

The revised manuscript addresses my key concern in the reversion by endopolygeny by showing plaques successfully form following washout. A more minor point is the introduction of division modes, which is not quite right. My key point is that the exogenous complementation in the UPRT locus is not a reversion experiment and actually contains very little information. At least the terminology should be revised, though removal of the experiment would be better.

Specific points

1. Introduction, lines 69-85 where schizogony, endodyogeny and endopolygeny are presented needs some clarification. Notably, endopolygeny is also internal budding in the cytoplasm, unlike schizogony. Furthermore, *Babesia* spp. divide by a binary form of schizogony, where nuclear and budding cycle are also tied together like in *Toxoplasma*, with the difference that they bud from the periphery rather than internally. The main distinction is internal vs external budding with the number of daughters ranging from 2 to many across both division strategies. See PMID: 32582569, DOI: 10.3389/fcimb.2020.00269
2. Fig 3. Looks like the association between centromere and outer-centrosome is lost upon 6 hr Aux treatment. If this is a representative image, please comment.
3. Figs 3B & S5B. How is it that a SFA2 per nucleus ratio of almost 1 informs that “TgSFA2 demonstrated that maturation of the centrosome might not be completed” whereas nearly the exact same ratio of Centrin per nucleus informs that absence of AP2-IX5 “does not have a drastic effect on the replication of the centrosome”. To me it looks like SFA2 and Centrin1 show the exact same thing, which predicts Cen1 and SFA2 are likely 100% overlapping. So the conclusion should be that centrosome maturation is normal, if SFA2 is indeed a marker for maturation (?). This is indeed concluded in the discussion!! (lines 632-634)
4. Fig S9. Complementation of AP2IX-5 in the UPRT locus is not a reversion experiment as an arrest is never induced in these parasites. On this point, I drastically differ in interpretation from the author, unless the complementation transfection was performed under continuous Aux pressure, which does not seem to be the case. This experiment really contains very little information and should be removed (my recommendation), but if it is decided not to delete this, the terminology used needs to be revised: it is not a “reversion” experiment; that is the experiment in Fig 6 and S10, S11, which is beautiful, so I really do not see the need for the complementation experiment in Fig S9.

Minor points

1. Figure S2. Please add the location of the diagnostic PCR primers used in panel B in panel A.
2. Fig S6B and S6C. please add Up- and Down-regulated in the figure, respectively, which makes is immediately interpretable.

Reviewer #2 (Remarks to the Author):

The authors have sufficiently addressed almost all my comments. But, I have some new suggestions on a few minor points as I indicated below.

In the title:

- I think the title should include TgAP2IX-5 since the whole research revolves around this key transcription factor.
- Also, I would refrain from using 'master'. Can I suggest for example: 'TgAP2IX-5 is a key transcriptional regulator of the asexual cell cycle division in *Toxoplasma gondii*'

In the introduction:

- Line 84: Regarding: "intermediate host of the parasite's lifecycle". I think 'employing several division patterns' also applies to the parasite developmental cycle in the definitive feline host where infection with tissue cysts containing bradyzoites advance to merozoites, schizonts and finally to sporocysts. So, not only in the intermediate host, but also in the definitive host, *Toxoplasma* can employ several division patterns. Therefore, I think "intermediate host of the parasite's lifecycle" should be replaced with "developmental stage of the parasite's lifecycle".

In the materials and methods:

- Line 132: Please introduce *Toxoplasma gondii* as RH strain when it is first mentioned in this section.
- Also, in the section titled "Parasite culture, transfection and purification," the source of reagents has been added, but I think the city and country of the suppliers should be also mentioned, unless this is not required by Nature Journals.
- Lines 163-164: "multiplicity of infection of 1.8 (MOI=1.80)" Could you clarify, normally multiplicity of infection is written as an integer, without fraction. It will help if the authors mention the parasite to host cell ratio.

Reviewer #3 (Remarks to the Author):

The authors have replied to most of my requests, however the quality of the figures has not really been improved. To cite only a few: there is still a disproportion between the size of the schematic for auxin degron strategy and the images of IFA, which are too small, while the main message for the reader is in the IFAs. The lettering of the IFA panel remains very small, compared to the rest of the text of fig. 1. Most figures still contain oversized parts next to parts which are too tiny. There is no homogeneity in the font between panels, so that some parts are difficult to read and others are really too big. These comments are true for most of the graphs. The zoom image of the EM picture is of very poor quality.

Minors :

-Line 70 : Should be *P. falciparum*

-Add the abbreviation of immunofluorescence assay after the first citation

-The following sentence « Interestingly, TgAP2IX-5 depletion induced the upregulation of 4 transcripts 433 encoding ApiAP2 transcription factors (AP2IX-5, AP2IX-1, AP2VI-3 and AP2X-9) (Table 434 S2). » would fit better line 493.

-To better highlight the possible negative feedback loop of TgAP2IX-5, I suggest to merge in a single paragraph (Line 512) the different results supporting this hypothesis (its upregulation in the presence of auxin and ChIP-seq peak of direct targeting).

-Fig. S8C, explain what ESS means in the legend of the figure

-I think it is overstated to say that TgAP2IX-5 controls the entire budding process (line 565). It might be only the initiation of the budding and once re-started the rest of the budding process is independent of TgAP2IX-5.

-line 571 : the number of nuclei per parasite increases in Fig. 6B with auxin time incubation, while it is written « decrease ».

Point by point response to reviewers.

Reviewer #1 (Remarks to the Author):

Summary

The revised manuscript addresses my key concern in the reversion by endopolygeny by showing plaques successfully form following washout. A more minor point is the introduction of division modes, which is not quite right. My key point is that the exogenous complementation in the UPRT locus is not a reversion experiment and actually contains very little information. At least the terminology should be revised, though removal of the experiment would be better.

Specific points

1. Introduction, lines 69-85 where schizogony, endodyogeny and endopolygeny are presented needs some clarification. Notably, endopolygeny is also internal budding in the cytoplasm, unlike schizogony. Furthermore, *Babesia* spp. divide by a binary form of schizogony, where nuclear and budding cycle are also tied together like in *Toxoplasma*, with the difference that they bud from the periphery rather than internally. The main distinction is internal vs external budding with the number of daughters ranging from 2 to many across both division strategies. See PMID: 32582569, DOI: 10.3389/fcimb.2020.00269

We clarified the part describing the budding in the different division modes and added the reference to *Babesia* schizogony. This is now stated in lines 77-79.

2. Fig 3. Looks like the association between centromere and outer-centrosome is lost upon 6 hr Aux treatment. If this is a representative image, please comment.

We did not find this to occur in the majority of case. This is now stated in the legend of figure 3.

3. Figs 3B & S5B. How is it that a SFA2 per nucleus ratio of almost 1 informs that “TgSFA2 demonstrated that maturation of the centrosome might not be completed” whereas nearly the exact same ratio of Centrin per nucleus informs that absence of AP2-IX5 “does not have a drastic effect on the replication of the centrosome”. To me it looks like SFA2 and Centrin1 show the exact same thing, which predicts Cen1 and SFA2 are likely 100% overlapping. So the conclusion should be that centrosome maturation is normal, if SFA2 is indeed a marker for maturation (?). This is indeed concluded in the discussion!! (lines 632-634)

We agree with the reviewer and changed the manuscript accordingly. This is now stated in lines 387-388.

4. Fig S9. Complementation of AP2IX-5 in the UPRT locus is not a reversion experiment as an arrest is never induced in these parasites. On this point, I drastically differ in interpretation from the author, unless the complementation transfection was performed under continuous Aux pressure, which does not seem to be the case. This experiment really contains very little information and should be removed (my recommendation), but if it is decided not to delete this, the terminology used needs to be revised: it is not a “reversion” experiment; that is the experiment in Fig 6 and S10, S11, which is beautiful, so I really do not see the need for the complementation experiment in Fig S9.

We agree with the terminology suggested by the reviewer and modified the text by removing all the terms referring to “reversion” in the complementation experiment. See lines 503-505 and 515-516.

Minor points

1. Figure S2. Please add the location of the diagnostic PCR primers used in panel B in panel A.

This was done as suggested by the reviewer.

2. Fig S6B and S6C. please add Up- and Down-regulated in the figure, respectively, which makes is immediately interpretable.

The figure S6B and S6C were changed as suggested by the reviewer.

Reviewer #2 (Remarks to the Author):

The authors have sufficiently addressed almost all my comments. But, I have some new suggestions on a few minor points as I indicated below.

In the title:

- I think the title should include TgAP2IX-5 since the whole research revolves around this key transcription factor.
- Also, I would refrain from using ‘master’. Can I suggest for example: ‘TgAP2IX-5 is a key transcriptional regulator of the asexual cell cycle division in *Toxoplasma gondii*’

The title was changed according to the reviewer comment.

In the introduction:

- Line 84: Regarding: “intermediate host of the parasite’s lifecycle”. I think ‘employing several division patterns’ also applies to the parasite developmental cycle in the definitive feline host where infection with tissue cysts containing bradyzoites advance to merozoites, schizonts and finally to sporocysts. So, not only in the intermediate host, but also in the definitive host, *Toxoplasma* can employ several division patterns. Therefore, I think “intermediate host of the parasite’s lifecycle” should be replaced with “developmental stage of the parasite’s lifecycle”.

The terms were changed according to the reviewer’ comment.

In the materials and methods:

- Line 132: Please introduce *Toxoplasma gondii* as RH strain when it is first mentioned in this section.

The RH $\Delta ku80$ Tir1 strain is now introduced in lines 123-130.

- Also, in the section titled “Parasite culture, transfection and purification,” the source of reagents has been added, but I think the city and country of the suppliers should be also mentioned, unless this is not required by Nature Journals.

Since this not a requirement, we did not add this information.

- Lines 163-164: “multiplicity of infection of 1.8 (MOI=1.80)” Could you clarify, normally multiplicity of infection is written as an integer, without fraction. It will help if the authors mention the parasite to host cell ratio.

We changed the sentence to state that the calculated parasite to host cell ratio was 1.8. See lines 155-156.

Reviewer #3 (Remarks to the Author):

The authors have replied to most of my requests, however the quality of the figures has not really been improved. To cite only a few: there is still a disproportion between the size of the schematic for auxin degron strategy and the images of IFA, which are too small, while the main message for the reader is in the IFAs. The lettering of the IFA panel remains very small, compared to the rest of the text of fig. 1. Most figures still contain oversized parts next to parts which are too tiny. There is no homogeneity in the font between panels, so that some parts are difficult to read and others are really too big. These comments are true for most of the graphs. The zoom image of the EM picture is of very poor quality.

We believe that the transformation to pdf severely altered the quality of our figures. We provide higher quality images as a tiff format. We changed the respective proportion of the figure 1A and 1B. We homogenized the font size according to the reviewer comments.

Minors :

-Line 70 : Should be *P. falciparum*

The text was changed according to the reviewer comment. See line 67.

-Add the abbreviation of immunofluorescence assay after the first citation

The text was changed according to the reviewer comment. See line 180.

-The following sentence « Interestingly, TgAP2IX-5 depletion induced the upregulation of 4 transcripts encoding ApiAP2 transcription factors (AP2IX-5, AP2IX-1, AP2VI-3 and AP2X-9) (Table 434 S2). » would fit better line 493.

The text was changed according to the reviewer comment. See line 476-480.

-To better highlight the possible negative feedback loop of TgAP2IX-5, I suggest to merge in a single paragraph (Line 512) the different results supporting this hypothesis (its upregulation in the presence of auxin and CHIP-seq peak of direct targeting).

The text was changed according to the reviewer comment. See lines 497-501.

-Fig. S8C, explain what ESS means in the legend of the figure

The text was changed according to the reviewer comment.

-I think it is overstated to say that TgAP2IX-5 controls the entire budding process (line 565). It might be only the initiation of the budding and once re-started the rest of the budding process is independent of TgAP2IX-5.

The text was changed according to the reviewer comment. See line 547.

-line 571 : the number of nuclei per parasite increases in Fig. 6B with auxin time incubation, while it is written « decrease ».

The text was changed according to the reviewer comment. See line 553.

Reviewer comments, third round -

No further comments.